# Intrathecal Injection of the Secretome from ALS Motor Neurons Regulated for miR-124 Expression Prevents Disease Outcomes in SOD1-G93A Mice

**DOI:** 10.3390/biomedicines10092120

**Published:** 2022-08-29

**Authors:** Marta Barbosa, Marta Santos, Nídia de Sousa, Sara Duarte-Silva, Ana Rita Vaz, António J. Salgado, Dora Brites

**Affiliations:** 1Faculty of Pharmacy, Research Institute for Medicines (iMed.ULisboa), Universidade de Lisboa, 1649-003 Lisbon, Portugal; 2School of Medicine, Life and Health Sciences Research Institute (ICVS), University of Minho, 4710-057 Braga, Portugal; 3ICVS/3B’s Associate Lab, PT Government Associated Lab, 4806-909 Guimarães, Portugal; 4Department of Pharmaceutical Sciences and Medicines, Faculty of Pharmacy, Universidade de Lisboa, 1649-003 Lisbon, Portugal

**Keywords:** ALS mouse model, anti-microRNA-124, intraspinal delivery route, neuroprotection, prevention of glial dysfunction, preservation of motor performance, secretome-based therapy, SOD1-G93A mutation

## Abstract

Amyotrophic lateral sclerosis (ALS) is a neurodegenerative disease with short life expectancy and no effective therapy. We previously identified upregulated miR-124 in NSC-34-motor neurons (MNs) expressing human SOD1-G93A (mSOD1) and established its implication in mSOD1 MN degeneration and glial cell activation. When anti-miR-124-treated mSOD1 MN (preconditioned) secretome was incubated in spinal cord organotypic cultures from symptomatic mSOD1 mice, the dysregulated homeostatic balance was circumvented. To decipher the therapeutic potential of such preconditioned secretome, we intrathecally injected it in mSOD1 mice at the early stage of the disease (12-week-old). Preconditioned secretome prevented motor impairment and was effective in counteracting muscle atrophy, glial reactivity/dysfunction, and the neurodegeneration of the symptomatic mSOD1 mice. Deficits in corticospinal function and gait abnormalities were precluded, and the loss of gastrocnemius muscle fiber area was avoided. At the molecular level, the preconditioned secretome enhanced NeuN mRNA/protein expression levels and the PSD-95/TREM2/IL-10/arginase 1/MBP/PLP genes, thus avoiding the neuronal/glial cell dysregulation that characterizes ALS mice. It also prevented upregulated GFAP/Cx43/S100B/vimentin and inflammatory-associated miRNAs, specifically miR-146a/miR-155/miR-21, which are displayed by symptomatic animals. Collectively, our study highlights the intrathecal administration of the secretome from anti-miR-124-treated mSOD1 MNs as a therapeutic strategy for halting/delaying disease progression in an ALS mouse model.

## 1. Introduction

Amyotrophic lateral sclerosis (ALS) is a fast progressive disease that results from the degeneration of the upper and lower motor neurons (MNs) in the motor cortex, brainstem, and spinal cord (SC). Consequently, this neurodegeneration leads to voluntary muscle weakness and wasting, resulting in a diversity of symptoms, such as dysarthria, dysphagia, weakness, and atrophy of the limbs. The pathological mechanisms underlying the disease include misfolded protein aggregation, mitochondrial dysfunction, dysregulated axonal transport, altered synaptic dynamics, excitotoxicity, and neuroinflammation [1]. It is nowadays accepted that the dysregulation of glial cells also contributes to the pathogenesis and progression of the disease [2]. Lately, the single-cell RNAseq of ALS microglia evidenced a molecular signature of the disease driven by the triggering receptor expressed on myeloid cells 2 (TREM2), defined as disease-associated microglia (DAM) [3]. By identifying DAM profiles, it is possible to better understand the heterogeneous microglia cell responses observed along the disease development. Currently, there is no effective treatment to prevent disease progression or change the disease course. One of the most commonly used models to explore ALS pathophysiology are mice bearing mutations [4]. Among these, superoxide dismutase 1 mice carrying the glycine to alanine point mutation at residue 93 (SOD1-G93A, mSOD1) recapitulate the symptoms of ALS patients and some of the glial phenotypes associated with the disease [5,6].

microRNAs (miRNAs) have been defined as important regulatory molecules for gene expression, and their dysregulation in ALS can be modulated toward the recovery of cell function and an increase in mSOD1 mouse lifespan, as reviewed in [7]. Furthermore, these molecules can be released into the secretome as free species or encapsulated in small extracellular vesicles (sEVs) [8,9,10,11] and internalised by neighbouring cells, also having the ability to influence distant cells and the extracellular environment [11,12]. Since ALS is characterized as a neuroinflammatory and neurodegenerative disease, we have focused on the study of inflammatory-associated (inflamma)-miRNAs, including miR-124, miR-155, and miR-146a, which were shown to be differentially expressed at the cellular and regional levels [12,13]. In particular, miR-124 is one of the most abundant miRNAs in the central nervous system and is involved in several important neuronal functions, such as neuronal differentiation/maturation and the regulation of synaptic activity [14]. However, we showed that its upregulation in in vitro models of ALS MNs was associated with neurodegeneration and that miR-124 disseminates through sEVs and causes time-dependent alterations in recipient microglial cells [11]. miRNA mimics and inhibitors have been proposed as therapeutics to modulate dysregulated miRNAs in cancer and multiple sclerosis [15,16]. Recently, we demonstrated that the incubation of the secretome from anti-miR-124-treated mSOD1 MNs (preconditioned secretome) was able to counteract the increased levels of IL-1β, IL-18, HMGB1, arginase 1, and inducible nitric oxide synthase (iNOS) in microglia treated with the mSOD1 secretome [13]. Interestingly, our data evidenced that the targeting of normal miR-124 values in ALS MNs enriched in miR-124 abrogates miR-125b overexpression and causes miR-146a/miR-21 downregulation in both cells and the secretome, thus inhibiting a pathological inflamma-miRNA profile in mSOD1 MNs. Such a preconditioned secretome also showed similar benefits in the spinal cord organotypic cultures (SCOCs) from mSOD1 mice by preventing the dysregulation of inflammation-associated genes and cell demise. This is not without precedent, since the secretome from mesenchymal stem cells has also been shown to exert neuroprotective, immunomodulatory, and regenerative effects in retinal degeneration, Parkinson’s disease, and SC injury, as examples [15,16,17]. Another study demonstrated the therapeutic potential of the administration of the secretome from adipose-derived stem cells in in vivo ALS mice by preventing MN loss and extending animal lifespan [18]. Importantly, conditioned media from human pluripotent stem cells showed neuroprotective effects on MNs, including those from ALS patients; improved neuromuscular junction; and delayed morbidity in mSOD1 mice [19]. The authors claimed that it has potential for autologous treatment in ALS. In addition, many studies have elucidated the therapeutic promise of miRNAs after their modulation, either by overexpression or suppression in mSOD1 mice and in astrocytes directly converted from patient somatic cells [8,20]. Some works have evidenced an increased lifespan of mice [20,21,22,23], followed by an improvement in MN survival [21,22] and muscle strength [21,23]. However, the benefits of the administration of the secretome derived from miRNA-modulated cells in in vivo ALS mice has never been explored.

For this reason, we decided to evaluate the therapeutic potential of anti-miR-124-treated mSOD1 MNs in preventing disabilities in mSOD1 mice. Thus, we performed an intrathecal injection of the preconditioned secretome in the mSOD1 mice at the early symptomatic disease stage. At the symptomatic stage, we evaluated the motor performance by testing the gait quality and the corticospinal function. Moreover, we collected the gastrocnemius muscle for muscle integrity evaluation and lumbar SC for neurodegeneration, neuroinflammation, and myelination assessments. Our results revealed that the preconditioned secretome prevented MN and glial pathological mechanisms and improved motor disabilities in the ALS mice. Therefore, this preconditioned secretome shows promise as a therapeutic tool to be tested in stratified patients with upregulated miR-124 levels, namely at symptom onset, contributing to the advance of precision medicine.

## 2. Materials and Methods

### 2.1. NSC-34 MN-like Cell Culture and Transfection Followed by sEV and Secretome Collection

NSC-34 MN cell line stably transfected with human WT SOD1 and human mSOD1 was grown in proliferation media (DMEM high-glucose with l-glutamine, no sodium pyruvate, supplemented with 10% FBS (Thermo Fisher Scientific, Waltham, MA, USA); 1% penicillin/streptomycin or pen/strep (Sigma-Aldrich, St. Louis, MO, USA); and 0.1% geneticin sulphate 418 (G418) for cell selection), as described in [13]. After 48 h of proliferation, differentiation was induced by changing the medium to DMEM-F12 plus 1% FBS; 1% nonessential amino acids (Merck, Darmstadt, Germany); 1% pen/strep; and 0.1% G418 [13]. After 24 h, mSOD1 MNs were transfected with 15 nM anti-miR-124 (Ambion^®^ Anti-miR^TM^ miRNA inhibitor, #AM10691) and mixed with X-tremeGENE™ HP DNA Transfection Reagent (Sigma-Aldrich, St. Louis, MO, USA) in a proportion of 2:1 and diluted in Opti-MEM™. Cells were left for 12 h, then fresh differentiation medium was added, and cells were maintained for an additional 48 h (4 DIV). This medium containing the factors secreted by mSOD1 MNs (secretome) was collected and centrifuged at 1000× *g* for 10 min to remove any cell debris. Since intrathecal injection requires very low volumes of the secretome [24], we concentrated it 100× using a Vivaspin™ 20 sample concentrator (5 kDa; GE Healthcare, Chicago, IL, USA). For this purpose, the secretome was subjected to a centrifugation of 4000× *g* at 4 °C for 180 min and then stored at −80 °C until being used in the intrathecal injection. In parallel and to confirm the successful entry of the injected secretome into the lumbar SC, we labelled the sEVs derived from the secretome of 4 DIV WT NSC-34 MNs and injected them intrathecally in the WT mice (as explained in Section 2.3). As previously described [12], we started to centrifugate the secretome at 1000× *g* for 10 min to remove cell debris. Then, we isolated the large EVs by centrifugation at 16,000× *g* for 1 h. The sEVs were further centrifuged in an Ultra L-XP100 centrifuge (Beckman Coulter Inc., Brea, CA, USA) at 100,000× *g* for 2 h. Finally, we labelled the sEVs with a PKH67 fluorescent linker kit (Sigma Aldrich, St. Louis, MO, USA) in accordance with the manufacturer’s specifications and resuspended them in DMEM-F12 plus 1% FBS depleted from the sEVs.

### 2.2. Transgenic mSOD1 Mouse Model

Mice were purchased from The Jackson Laboratory (Bar Harbor, ME, USA), namely transgenic B6SJL-TgN mSOD1Gur/J males (no. 002726) and non-transgenic B6SJLF1/J wild-type (WT) females. They were housed at the animal facility of the Life and Health Science Research Institute (ICVS)*,* University of Minho, where the colony was also established. They were maintained under standard conditions (12 h light/12 h dark cycles, room temperature (RT) at 22–24 °C, and 55% humidity) and provided with food and water ad libitum. The colony was maintained on a background B6SJL by breeding mSOD1 transgenic males with non-transgenic females. Transgenic mSOD1 mice were compared to aged-matched WT mice. The procedures performed were in accordance with the European Community guidelines (Directives 86/609/EU and 2010/63/EU, Recommendation 2007/526/CE, European Convention for the Protection of Vertebrate Animals used for Experimental or Other Scientific Purposes ETS 123 (https://rm.coe.int/168007a445 accessed on 1 June 2022) and the Portuguese Laws on Animal Care (Decreto-Lei 129/92, Portaria 1005/92, Portaria 466/95, Decreto-Lei 197/96, Portaria 1131/97). All the protocols used in this study were approved by the Portuguese National Authority (General Direction of Veterinary) and Ethical Subcommittee in Life and Health Sciences (SECVS; ID: 018/2019).

### 2.3. Intrathecal Injection of the sEVs and Secretome

The sequence of the experiments developed in this study is graphically summarized in Figure 1. The control groups used in the experiment were the WT and mSOD1 mice injected with the vehicle (differentiated NSC-34 cell media with 1% FBS and without sEVs), as used by others [25,26], with a total of 8 and 7 animals, respectively. The choice of such controls was based on our previous studies, whereby we evidenced that: (1) the secretome from cultured WT cells induced a decrease in several biomarkers associated with inflammatory status and phagocytic and synaptic genes in the SCOCs of early symptomatic mSOD1 mice (unpublished data); (2) the secretome from mSOD1 MNs caused cell demise in both WT and mSOD1 SCOCs [13]; and (3) the secretome from mSOD1 MNs activated the polarization of WT microglia (published in [13]). Therefore, we chose to use the MN cell culture media injection (vehicle) to not interfere with the natural life progression in the WT animals or the disease progression from the early stage to the symptomatic stage in the mSOD1 mice. By adopting this approach, we could document the behavioural and molecular disease progression in the transgenic mice relative to the WT animals. Such conditions are requisite for drawing conclusions on the potential benefits of using the preconditioned secretome from anti-miR-124-treated ALS MNs. Our final aim is the translation of such a strategy for autologous application in ALS patients showing upregulated levels of miR-124 after a stratification assessment. In parallel experiments, to assess the permanence of the secretome sEV components in the SC after their injection into the WT mice, we used the sEVs isolated from the secretome labelled with the PKH67 fluorescent cell linker kit, as described above.

Experiments to assess the therapeutic efficacy of the anti-miR-124-treated mSOD1 MNs were performed in the mSOD1 mice injected with the concentrated preconditioned secretome. The 12-week-old animals (early symptomatic stage of the disease) were anesthetized intraperitoneally (i.p.) with ketamine (75 mg/kg) plus medetomidine (0.5 mg/kg) [27]. Once anesthetized, we proceeded to the single intrathecal injection. WT and mSOD1 mice were firmly held by a pelvic girdle, which was in line with the sixth lumbar vertebral body (L6), as described previously [28]. Then, we identified the gap between the L4 and L5, located above the ileac crest, by palpation through the skin and inserted the needle at the midspinal line [28]. We used a 30-gauge needle Hamilton syringe (Hamilton, Bonaduz, Switzerland) to slowly inject the sEVs, vehicle, or secretome in a proportion of 1 µL per gram of animal weight [24]. The successful entry into the lumbar cistern was confirmed with a sudden tail flick after the needle insertion. Then, the animals were injected i.p. with atipamezole hydrochloride (1 mg/mL; Antisedan^®^, Pfizer, Inc., Brooklyn, NY, USA) for the reversal of the anesthesia [29]. The WT animals injected with sEVs derived from WT MNs were sacrificed at 8 h and 72 h post injection, based on our previous findings in microglia showing sEV engulfment after 24 h of incubation [11] and the results of other studies demonstrating the distribution of labelled sEVs in the SC 1 week after intrathecal injection [25]. Additional studies in other conditions are unanimous in considering the presence of sEVs at 72 h after their injection [26,30]. Then, we decided to explore the presence of labelled sEVs in the lumbar SC to determine their permanence using immunohistochemistry in the collected samples at 8 h and 72 h post injection, corresponding to acute and lasting distribution.

Note that no statistical differences were found in the mouse weight (g) between WT and transgenic animals, before or after treatment (WT + vehicle, 25.9 ± 1.0 and 24.9 ± 1.4; mSOD1 + vehicle, 24.6 ± 0.9 and 25.2 ± 1.3; mSOD1 + secretome, 25.9 ± 0.5 and 26.8 ± 1.2).

### 2.4. Behavioural Tests

To determine the impact of the secretome from anti-miR-124-treated mSOD1 MNs on motor behaviour, we evaluated motor performance in the animals two weeks after the injection. We assessed the gait quality by the footprint test, the muscular strength by the hanging wire test, and the spontaneous activity by the cylinder test. Finally, the deficits in the corticospinal function were also examined using the limb clasping and grasping tests.

#### 2.4.1. Footprint Test

The fore and hind paws of mice were painted with non-toxic dyes of different colours, and the mice were placed on absorbent paper in an inclined corridor so that they would walk up it since mice have the tendency to run upwards to escape [29]. We measured the stride length (in centimeters), which is the distance between the center of the fore-foot plantar and the center of the hind-foot plantar on the same side of the body, within the same stride [29]. A shorter stride length indicated abnormalities in the gait.

#### 2.4.2. Hanging Wire Grid Test

The mice had to remain clinging to an inverted surface of a cage lid, demonstrating their grip strength [29]. The duration (in seconds) for which the mice remained grasping the cage was measured until it reached 120 s [29].

#### 2.4.3. Cylinder Test

The mice were placed into a clear cylinder, and the number of times they reared up against the cylinder wall [32] was counted for 3 min. This test demonstrates animals’ forelimb activity and their spontaneous exploration of the environment [32].

#### 2.4.4. Clasping and Grasping Tests

The mice were suspended by holding their tails, and the position of their hindlimbs and toes was observed for several seconds. An abnormal phenotype is represented by the retraction of the hindlimbs toward the abdomen (clasping reflex) [29] and/or the curling of the toes (grasping reflex) during the suspension time, indicating the presence of motor dysfunctionality.

### 2.5. Homogenates and Tissue Slices

To avoid the animal suffering associated with the disease phenotype, we sacrificed the animals at 15 weeks of age (one week after the beginning of the behaviour tests). Mice were anesthetized i.p. with ketamine (75 mg/kg) plus medetomidine (0.5 mg/kg) and transcardially perfused with 0.1 M PBS at pH 7.4. Then, lumbar SC and left gastrocnemius muscle were dissected and rapidly frozen at −80 °C for reverse transcription quantitative real-time polymerase chain reaction (RT-qPCR) and western blot (WB) assays. For histological and immunohistochemistry studies, animals were perfused with a fixative containing 4% paraformaldehyde in PBS, and the lumbar SC and left gastrocnemius muscle were removed. Then, the lumbar SC was post-fixed with the same fixative for 24 h at 4 °C and preserved in 30% sucrose solution. The gastrocnemius muscle was stretched over a notched wooden stick to maintain the in vivo length and to prevent over-contraction. Later, the muscle was frozen in isopentane cooled over liquid nitrogen and stored at −80 °C [33].

For sectioning, the muscle was sliced directly in the cryostat, while the SCs were first embedded in Tissue-Tek^®^ O.C.T. Compound (Sakura Finetek-USA, Torrance, CA, USA) before transversal 20 μm thick sections were serially cut using the cryostat Leica CM1900 (UV Leica; Wetzlar, Germany). The sections were collected on Superfrost Plus glass slides (Thermo Scientific, Waltham, MA, USA) and preserved at −80 °C.

### 2.6. RT-qPCR

The lumbar SC and muscle were homogenized in TRIzol reagent (Invitrogen, Waltham, MA, USA) using a Pellet Mixer (VWR Life Science, EUA). The muscle homogenization in TRIzol reagent was performed by cutting the tissue into small pieces, followed by transferring the resulting mixture through a 1 mL syringe with a 20 G needle until a homogeneous mixture was obtained. The total RNA was extracted and quantified on a NanoDrop ND100 Spectrophotometer (NanoDrop Technologies, Wilmington, DE, USA) according to the standard procedure in our laboratory [12].

For gene expression, the total RNA was reverse transcribed into cDNA using the Xpert cDNA Synthesis Supermix kit (GRiSP, Porto, Portugal). RT-qPCR was performed using an Xpert Fast SYBR Mastermix BLUE kit (GRiSP) and the primer sequences listed in Appendix A. The following conditions were used for each amplification product: 50 °C for 2 min, 95 °C for 2 min, followed by 40 amplification cycles at 95 °C for 5 s and 62 °C for 30 s. The amplification cycles for our samples were no higher than 30. The specificity of the amplified product was verified by a melting curve. The ribosomal protein L19 (RPL19) was used as the endogenous control to normalize each gene expression level.

For miRNA expression, cDNA was performed using a miRCURY LNA^TM^ RT Kit (QIAGEN, Valencia, CA, USA) and RT-qPCR using PowerUp™ SYBR™ Green Master Mix (Applied Biosystems, Life Technologies, Waltham, MA, USA) and the primer sequences listed in Appendix A. The operating conditions were 95 °C for 10 min, followed by 50 amplification cycles at 95 °C for 10 s and 60 °C for 1 min (ramp rate of 1.6°/s). Our results showed amplification cycles no higher than 40. At the end, a melting curve analysis was carried out to verify the specificity of the amplified product. SNORD110 was used as the endogenous control.

The expression of mRNA/miRNA was measured via the 2^−ΔΔCt^ method relative to that of the endogenous control. Each sample was measured in duplicate. The cDNA synthesis was performed in a thermocycler (Biometra^®^, Göttingen, Germany), and the RT-qPCR was run on a QuantStudio 7 Flex Real-Time PCR System (Applied Biosystems).

### 2.7. Histology and Immunohistochemistry

The gastrocnemius muscle was stained with hematoxylin and eosin using an AutoStainer XL (Leica). Briefly, the muscle sections were rinsed in distilled water and placed into Harris hematoxylin solution for 1 min, followed by a tap-water wash for 2 min. Then, differentiation with 0.5% ammonia water was performed for 10 s. After washing with tap water for 2 min, the sections were stained with eosin for 20 s; dehydrated using 70, 95, and absolute alcohol for 2 min each; and covered with xylol for 1 min. The samples were mounted with a glass coverslip using Entellan mounting media (Merck Millipore, Burlington, MA, USA). Photos were obtained using an Olympus widefield inverted microscope IX53 with 20× magnification. The images from three fields of the same muscle section per animal (from three animals per group) were analysed using Fiji software. We measured the mean area occupied by each fiber from the transversal section of the muscle.

To evaluate neurodegeneration, the lumbar SC was stained with Fluoro-Jade B (Chemicon International, Temecula, CA, USA). The samples were defrosted at RT for 10 min. Then, they were incubated in 0.06% potassium permanganate solution for 30 min with gentle agitation. After a 5 min PBS rinse, the samples were stained with a 0.001% solution of Fluoro-Jade B dissolved in 0.1% acetic acid vehicle for 30 min with mild agitation. The samples were then rinsed through three changes of PBS for 3 min each with agitation and mounted in Fluromount-G (Sigma-Aldrich) with a glass coverslip on the top.

To access the distribution of sEVs in the SC, we also stained the neurons and astrocytes with NeuN and glial fibrillary acidic protein (GFAP), respectively. The presence of glial cells in the lumbar SC was also evaluated through the quantification of ionized calcium-binding adaptor molecule 1 (Iba-1) for microglia and GFAP for astrocytes. The previously described protocol was followed with minor modifications [13]. After defrosting, the permeabilization/blocking of the sections was performed using Hank’s balanced salt solution with 2% heat-inactivated horse serum, 10% FBS, 1% BSA, 0.25% Triton X-100, and 1 nM HEPES for 3 h at RT. Then, the sections were incubated for 48 h with the primary antibodies (indicated in Appendix A) at 4 °C. Following 3 washes with 0.01% Triton-X in PBS for 20 min, the sections were incubated for 2 h at RT with secondary antibodies (indicated in Appendix A). The cell nucleus was stained with 4′,6-diamidino-2-phenylindole (DAPI) for Neu N/GFAP staining and Hoechst dye for Iba-1/GFAP labelling. Both solutions were diluted in PBS (0.1 µg/mL) for 10 min. Finally, the sections were mounted in Flouromount-G with a glass coverslip.

The fluorescence images were obtained in a Leica DMi8-CS inverted microscope with Leica LAS X software, and the different z-stacks were merged and analysed with Fiji software. We measured the mean fluorescence of the ventral horn of one lumbar SC transversal section per animal (N = 3 per group) stained with Fluoro-Jade B and obtained with 20× magnification. Images stained with Iba-1 and GFAP were obtained with 40× magnification and analysed from five ventral horn fields per animal (from three animals per group). We measured the fraction of the area occupied by the GFAP- and Iba-1-positive cells.

### 2.8. Western Blot

Total protein was isolated from the organic phase of the TRIzol–chloroform from lumbar SC and muscle [34] and quantified using a Bradford protein assay kit. The protein expression was assessed by WB analysis according to the standard procedure in our laboratory [35]. Equivalent amounts of protein were separated on dodecyl sulfate polyacrylamide gel electrophoresis (SDS-PAGE) and transferred to a nitrocellulose membrane for 1 h 30. The membranes were blocked for 1 h with 5% non-fat milk in TBS-T (0.1% Tween-20), followed by overnight incubation with primary antibodies (listed in Appendix A) and gentle agitation. Membranes were then washed 3 times, for 5 min, with TBS-T and incubated with secondary antibodies conjugated with horseradish peroxidase (listed in Appendix A) for 1 h at RT with mild agitation. Immunoreactive bands were detected after the incubation with Western Bright™ Sirius (K-12043-D10, Advansta, Menlo Park, CA, USA) using an iBright^TM^ FL1500 Imaging System. Bands were quantified using iBright^TM^ analysis software. Results were normalized to the expression of β-actin and indicated as fold change.

### 2.9. Statistical Analysis

Non-parametric ANOVA (Kruskal–Wallis test) and unpaired and non-parametric Mann–Whitney tests were used when the data showed non-normal distribution. Non-continuous categorical variables were analysed using the chi-square (and Fisher’s exact) test. For data with normal distribution (Shapiro–Wilk test *p* > 0.05), one-way ANOVA was applied, followed by multiple-comparisons Bonferroni post hoc correction, as well as unpaired and parametric Student’s *t*-tests. Welch’s *t*-test correction was applied when variances were different between the two groups. Values of *p* < 0.05 were considered statistically significant. Data were expressed as mean ± SEM values, except for the non-continuous categorical variables. Results were analysed using GraphPad Prism 8.0.1 (GraphPad Software, San Diego, CA, USA).

## 3. Results

### 3.1. sEVs Administered by Intrathecal Injection in B6SJLF1/J Wild-Type (WT) Mice Are Identified in Lumbar SC Slices after 8 and 72 h of Delivery in 12-Week-Old Animals

We previously demonstrated the upregulation of miR-124 in mSOD1 NSC-34 MNs and its recapitulation by the sEV-free secretome and sEVs, which caused alterations in the microglia phenotype once engulfed [11]. Increased miR-124 levels were also observed in the SC of mSOD1 mice at the symptomatic stage [36]. Considering that the regulation of miR-124 in MNs could have neuroprotective effects, we later demonstrated that the secretome from anti-miR-124-treated mSOD1 MNs prevented microglia activation and SC pathogenicity relative to the secretome from non-treated mSOD1 MNs [13]. Thus, in the present study, we tested whether the injection of the preconditioned secretome could halt or delay disease progression in an mSOD1 mouse model.

To gain an understanding of the permanence of the sEVs after being injected as components of the secretome, we labelled the sEVs isolated from the secretome of WT NSC-34 MNs with PKH67 fluorescent cell linker before their intrathecal injection in the 12-week-old WT mice, as explained in Section 2.3 of the Materials and Methods. We assessed their presence in the collected lumbar sections after short (8 h) and long (72 h) periods of time.

We were able to identify sEVs (in green) after the intrathecal injection for both periods of time (Figure 2A,B). To further assess their distribution among neurons and astrocytes, we performed immunohistochemical staining for NeuN (Figure 2C) and GFAP (Figure 2D), respectively, using DAPI for nuclei and PKH67 for sEV labelling. sEVs were identified in both cell images at 72 h after injection, demonstrating that they could be disseminated by the whole secretome and mediate lasting paracrine signalling with neural cells.

From here on we decided to use the whole secretome, and not only the injection of sEVs, considering its advantages for ALS therapeutics. On the one hand, storing and administering the secretome is easier than doing so for sEVs and facilitates their preservation. On the other hand, the secretome also contains additional bioactive molecules that may increase its neuroprotective potential, as a consequence of the effects of anti-miR-124 on MN survival [13]. When considering ALS patients, this study may also open up new possibilities regarding the manipulation and engineering of their cells toward the collection of the secretome for autologous transplantation.

### 3.2. Expression of miR-124 in the SC of mSOD1 Mice Is Downregulated after 3 Weeks of Intrathecal Injection of Secretome from Anti-miR-124-Treated mSOD1 MNs

In our prior studies, we demonstrated that miR-124 levels in the secretome recapitulate the levels observed in cells, either non-modulated or treated with its mimics and inhibitors [9,10,11,13]. Here, we intended to test whether the intrathecal injection of the secretome from anti-miR-124 mSOD1 MNs in pre-symptomatic mSOD1 mice was able to reduce the miR-124 expression in the lumbar SC of ALS mice when assessed at 15 weeks of age. This was important for inferring the potential beneficial effects that the approach may have.

As depicted in Figure 3A and Appendix A, the expression of miR-124 was found to be upregulated in the mSOD1 mice treated with the vehicle, as compared with the matched WT animals (*p* < 0.01). The injection of the secretome from anti-miR-124-treated MNs, under the same conditions, was successful in downregulating the expression of miR-124 (Figure 3B, *p* < 0.05 and Appendix A).

In the next sections, we will explore the efficacy of this novel strategy in inhibiting the progression of the disease in mSOD1 mice from the early symptomatic stage to the symptomatic stage by preventing motor disabilities, as well as neurodegeneration, neuroinflammation, and demyelination.

### 3.3. Motor Disabilities in Transgenic Mice at the Early Symptomatic Stage of ALS Are Reversed by Intrathecal Injection of Secretome from Anti-miR-124-Treated ALS MNs

As already mentioned, the downregulation of miR-124 in ALS MNs was shown to be neuroprotective and to regulate neuroinflammation in in vitro and ex vivo experimental ALS models [13]. However, this has never been tested in ALS in vivo models, such as mSOD1 mice. We started by assessing the behavioural alterations in the mSOD1 mice at the symptomatic stage (14-week-old mice), as the deterioration of motor performance at this stage of the disease was previously observed in an ALS mice model [37,38,39,40,41,42,43,44].

As indicated in Figure 4, we observed a decrease in the stride length, the delay before falling from the grid, and the number of times that the animal reared up and touched the cylinder for the vehicle-treated transgenic mice, in comparison with matched WT animals (Figure 4E–G, *p* < 0.05). These observations are associated with an abnormal gait quality, muscle weakness, and less spontaneous vertical activity, respectively. In addition, a large percentage of mSOD1 animals showed enhanced limb clasping (Figure 4H, *p* < 0.01 vs. WT + vehicle) and grasping (Figure 4I, *p* < 0.0001 vs. WT + vehicle) reflexes, in accordance with abnormalities in the corticospinal tract.

The secretome from anti-miR-124-treated mSOD1 MNs was able to inhibit such impairments. We noticed improvements in all the assessed behavioural tests (Figure 4E–G), though only significant for the stride length, i.e., the distance from the heel print of one foot to the heel print of the other foot during the walking test, evidencing the amelioration of the gait quality, muscle strength, and spontaneous activity. Moreover, the reduction in the percentage of animals with dystonic movements associated with clasping (Figure 4H, *p* < 0.0001 vs. mSOD1 + vehicle) and grasping reflexes (Figure 4I, *p* < 0.01 vs. mSOD1 + vehicle) was similarly clear 2 weeks after the treatment with the secretome from anti-miR-124-treated mSOD1 MNs.

### 3.4. Atrophy of Gastrocnemius Muscle in Early Symptomatic ALS Mice, as Well as Loss of Motor Neuron Viability and Synaptic Dynamics, Are Reversed by Intrathecal Injection of the Secretome from Anti-miR-124-mSOD1 MNs

Since we observed motor performance disabilities in the mSOD1 mice, a common symptom of ALS caused by muscle weakness [45], already described in SOD1 models of ALS [46], and their rescue by the intrathecal injection of the preconditioning secretome, we next investigated gastrocnemius muscle fibers in the absence of and after treatment using hematoxylin and eosin staining. The measurement of the cross-sectional areas revealed thinner muscle fibers in the transgenic than in the WT mice (*p* < 0.0001 vs. WT + vehicle, Figure 5A,B). We additionally identified the decreased gene expression of hexaribonucleotide binding protein 3 (NeuN), presynaptic synaptophysin, and postsynaptic density protein 95/PSD-95 (at least *p* < 0.05 vs. WT + vehicle, Figure 5C and Appendix A), suggesting the existence of neuronal dysfunction and the eventual compromise of neuromuscular transmission and muscle functionality.

Notably, 3 weeks after the intrathecal injection of the preconditioning secretome from anti-miR-124 mSOD1 MNs in the ALS mice, we noticed the reversal of the skeletal muscle atrophy (*p* < 0.01 vs. mSOD1 + vehicle, Figure 5A,B), as well as of neuronal demise and deregulated synaptic activity (at least *p* < 0.05 vs. mSOD1 + vehicle, Figure 5D and Appendix A).

The data highlighted the efficacy of the anti-miR-124 preconditioned secretome as a promising therapy for skeletal muscle remodelling in ALS with potential to be translated to patients.

### 3.5. Neurodegeneration in the Lumbar SC of Early-Symptomatic ALS Mice Is Prevented after Injection of Secretome from Anti-miR-124-Treated mSOD1 MNs

In previous studies, we observed the existence of neurodegeneration in the SC of mSOD1 mice [36] and that the downregulation of miR-124 restored mSOD1 MN viability toward control levels by preventing early apoptosis [13]. Based on the above results, we decided to explore the ability of the preconditioned secretome to prevent neurodegeneration when injected in 12-week-old mSOD1 mice. For this, we collected the lumbar SC at 3 weeks after the intrathecal administration and stained the transversal sections with Fluoro-Jade B. As expected, we found a higher fluorescence intensity in the SC from mSOD1 mice at 15 weeks of age than in the matched WT animals (*p* < 0.01, Figure 6A,B), demonstrating an increased number of degenerating neurons. In contrast, in the samples obtained from the mSOD1 mice treated at 12 weeks of age with the secretome from anti-miR-124-treated mSOD1 MNs, we noticed a decrease in Fluoro-Jade B-positive staining relative to the vehicle-treated mSOD1 mice (*p* < 0.05, Figure 6A,B), demonstrating the secretome’s preventive effect against neurodegeneration in the ALS animals.

### 3.6. MN Loss and Dysfunction, as Well as Myelination Impairment, Are Averted in the Lumbar SC of ALS Mice Injected with the Secretome from Anti-miR-124-Treated mSOD1 MNs

Given our results suggesting the neuroprotective potential of the secretome from anti-miR-124-transfected mSOD1 MNs, we further explored the markers related to MN functionality. As shown in Figure 7A and Appendix A, we first observed the decreased gene expression of PSD-95, dynein, kinesin, and C-X3-C motif chemokine ligand 1 (CX3CL1, fractalkine), which accompanied that of NeuN (at least *p* < 0.05 vs. WT + vehicle) when assessed in the lumbar SC of the mSOD1 mice at 15-week-old. These results revealed an impairment of the postsynaptic dynamics, retrograde/anterograde axonal transport, and CX3CL1 that may have compromised paracrine signalling with the microglia receptor CX3CR1. Neurodegeneration was further confirmed by the decrease in the protein NeuN (*p* < 0.01 vs. WT + vehicle, Figure 7B and Appendix A). The data indicated that the postsynaptic deregulation and neuronal loss at the gastrocnemius muscle/neuromuscular junction was also present in the SC of the mSOD1 mice. It should also be noted that despite the alterations in PSD-95, we did not detect changes in the gene encoding for the presynaptic synaptophysin protein in the SC of the ALS animals.

Neuronal survival and axonal preservation depend on myelin integrity [47], an issue that has scarcely been assessed in ALS patients and experimental models. Here, we assessed the mRNA levels of the myelin basic protein (MBP) and the myelin proteolipid protein (PLP), which were revealed to be downregulated in the symptomatic mSOD1 SCs (at least *p* < 0.05 vs. WT + vehicle, Figure 7C and Appendix A). G protein-coupled receptor 17 (GPR17) mRNA, an important regulator of oligodendrocyte precursor (OPC) maturation, was previously found to be increased in mSOD1 mice [48]. We confirmed a 2.5-fold increase in its gene expression in these transgenic animals (*p* < 0.01).

The intrathecal injection of the preconditioned secretome exerted protective effects against most of the observed impairments. In fact, we observed an upregulation of PSD-95 and NeuN (including the protein expression) (*p* < 0.05 vs. mSOD1 + vehicle, Figure 7D,E and Appendix A), accompanied by an increase in MBP and PLP mRNA levels (at least *p* < 0.05 vs. mSOD1 + vehicle, Figure 7F and Appendix A), highlighting the benefits of the tested secretome-based strategy in avoiding demyelination.

These results call our attention to the efficacy of the conditioned secretome in halting disease progression in early symptomatic mSOD1 mice.

### 3.7. Intrathecal Injection of the Secretome from Anti-miR-124-Treated mSOD1 MNs in 12-Week-Old mSOD1 Mice Counteracts Lumbar SC Immune Deregulation at the Symptomatic Stage

Neuroinflammation is associated with ALS, and glial cell dysregulation is implicated in the initiation/propagation of neurodegeneration [49,50]. Here, based on the promising data obtained for neuronal function with the intrathecal injection of the preconditioned secretome in 12-week-old mSOD1 mice, we wondered whether such beneficial effects could be supported by the prevention of the homeostatic imbalance of microglia and astrocyte activation.

Concerning the categorization of microglia immunoreactivity, we found the downregulation of arginase 1 in the mSOD1 SC 3 weeks after vehicle injection in the early symptomatic mice (Figure 8A, *p* < 0.05 vs. WT + vehicle), known to exacerbate neuropathological and behavioural deficits [51], as well as the gene purinergic receptor p2y12/P2RY12 (*p* < 0.01 vs. WT + vehicle, Figure 8A and Appendix A), which may compromise microglial motility and migration [52]. In addition, we identified low levels of TREM2 (*p* < 0.001 vs. WT + vehicle) and the milk fat globule epidermal growth factor 8 (MFG-E8) (*p* < 0.05 vs. WT + vehicle), which have been shown to compromise phagocytic ability [3,53,54]. Notably, the significant changes observed in arginase 1 and MFG-E8 in mSOD1 SC + vehicle compared only with WT SC + vehicle were not observed when comparing the three conditions together (Appendix A). The induced regulation of microglial activation may have been determined by the upregulated levels of CX3CR1, TMEM119, and TIMP2 (at least *p* < 0.05 vs. WT + vehicle), which have been shown to regulate neuroinflammation [55,56,57]. An inverted signature was obtained when we assessed the SC parameters 3 weeks after the intrathecal injection of the preconditioned secretome, namely those related to microglia phagocytosis (e.g., TREM2, *p* < 0.05 vs. mSOD1 + vehicle, Figure 8D and Appendix A) and arginase 1 (*p* < 0.05 vs. mSOD1 + vehicle), which are associated with a reparative phenotype.

Regarding the reactive profile of astrocytes, Figure 8B and Appendix A show increased levels of GFAP (*p* < 0.05 vs. WT + vehicle, Figure 8B) and connexin-43/Cx43 (*p* < 0.001 vs. WT + vehicle, Figure 8B) in the lumbar SC of mSOD1 mice, pointing to the existence of reactive astrocytes.

Interestingly, the preconditioned secretome led to a significant reduction in these parameters (GFAP, *p* < 0.01 and Cx43, *p* < 0.05 vs. mSOD1 + vehicle). Note the non-existence of alterations in the gene expression of S100 calcium-binding protein B (S100B) in the absence or presence of the preconditioned secretome.

When we assessed the dysregulation of genes associated with inflammation, we observed an imbalance with decreased iNOS and interleukin (IL)-10, together with elevated levels of tumour necrosis α/TNF-α (at least *p* < 0.01 vs. WT + vehicle, Figure 8C and Appendix A), while no changes were observed for IL-1β. The secretome caused an elevation in IL-10 and iNOS (*p* < 0.05 vs. mSOD1 + vehicle), which may have compensated for each other to reach a steady state. Overall, our results highlighted that the preconditioned secretome regulated the lumbar SC immunoreactivity.

We also evaluated the microglial and astrocytic prevalence in the slices by assessing the area stained for Iba-1 and GFAP, respectively. An increase in the area occupied by GFAP- and Iba-1-positive cells was found in the mSOD1 SC (*p* < 0.01, Figure 9C,D, respectively) in comparison to the matched WT samples. The expression of proteins as assessed by WB analysis also indicated an elevation in GFAP and S100B (though not significant) and vimentin and Iba-1 in mSOD1 SC, in comparison with the WT samples (Figure 9E,F; *p* < 0.05 and Appendix A), corroborating the astrocyte reactivity and microglia activation. Interestingly, the modulated secretome not only decreased the area occupied by GFAP- and Iba-1-positive cells (at least *p* < 0.05 vs. mSOD1 + vehicle, Figure 9C,D), but also the protein levels of GFAP, S100B, and Iba-1 (*p* < 0.05, Figure 9G,H) in comparison with the SC from mSOD1 mice injected with the vehicle, supporting the immune remodelling ability of the preconditioned secretome.

### 3.8. The Secretome from Anti-miR-124-Treated mSOD1 MNs Counteracts the Upregulation of Inflamma-miRNAs in the Lumbar SC of mSOD1 Mice

Several miRNAs have been found to be dysregulated in ALS patients [58], as well as in cortical brain and SC samples from mSOD1 mice [36,59], including miR-155, miR-146a, miR-21, and miR-125b. Therefore, and based on the obtained data, we wondered whether the preconditioned secretome would be able to control such miRNAs, considering the reduction in anti-miR-124 levels observed in the SC of mSOD1 mice treated with the preconditioned secretome (Figure 3).

Firstly, we were interested in confirming that such miRNAs were upregulated in the mSOD1 mice treated with the vehicle at 12-week-old and analysed 3 weeks thereafter, the design we used to test the potential therapeutic benefits of the intrathecal injection of the preconditioned secretome from the anti-miR-124 treated mSOD1 MNs.

As expected, all the assayed miRNAs, except miR-125b, were found to be upregulated in the SC tissue collected from the mSOD1 mice at 15-week-old, corresponding to the symptomatic stage (Figure 10A, at least *p* < 0.05 vs. WT + vehicle and Appendix A).

Interestingly, the preconditioned secretome was able to counteract the upregulation of the 3 main inflamma-miRNAs (Figure 10B and Appendix A), i.e., miR-146a (*p* < 0.05), miR-155 (*p* < 0.01), and miR-124 (*p* < 0.05), when compared with the mSOD1 mice injected with the vehicle (Figure 10A), attesting to its regulatory potential in the prevention of inflamma-miRNA deregulation.

In sum, our results highlight the therapeutic potential of the secretome from anti-miR-124-modulated mSOD1 MNs in preventing motor disabilities, muscular atrophy, MN degeneration/demise/dysfunctionality, glial reactivity, impaired phagocytosis, and inflammatory imbalance. We hope to translate this therapeutic strategy to ALS patients, mainly to those with upregulated miR-124, to slow down disease progression and prolong their lifetimes.

## 4. Discussion

In the present study, our main interest was to validate whether the benefits of regulating miR-124 in mSOD1 MNs using anti-miR-124 were translated to mSOD1 mice in terms of preventing disease progression. We showed that upregulated levels of miR-124 in mSOD1 MNs [11] were associated with neurodegeneration and that its secretome caused glial activation, findings that were preserved when we transfected mSOD1 MNs with anti-miR-124 [13].

miR-124 is the most abundant miRNA in the adult brain and has been described as a regulator of microglia-mediated inflammation in neurological diseases [10,60]. It is mainly expressed in neuronal cells, showing important functions in neuronal development, differentiation, synaptic plasticity, and even in memory signalling molecules [61]. The elevated expression of miR-124 in Alzheimer’s disease (AD) was shown to benefit the neurite outgrowth, mitochondria activation, small Aβ oligomer reduction, and beta-site amyloid precursor protein cleaving enzyme 1 (BACE1) in experimental models and patients [9,10,62]. However, in ALS, the upregulation of miR-124 in mSOD1 MNs was shown to impair the neurite network, mitochondria dynamics, axonal transport, and synaptic signalling [13]. Moreover, the secretome from these ALS MNs also revealed elevated levels of miR-124 and induced pathological features in the SC organotypic cultures from early symptomatic ALS mice [11,13]. The beneficial or harmful effects of miR-124 may then depend on the cell type and the pathological condition.

miRNA mimics and inhibitors have been used to change the endogenous levels of certain miRNAs and regulate cell function [63,64], as in the case of miR-124 [65,66]. The overexpression of miR-124 was shown to act as a neuroprotective factor in Parkinson’s disease [67,68] and in AD [62,69]. In human AD neuronal models with elevated levels of miR-124, we have shown by using an miR-124 mimic and inhibitor that an increase in miR-124 may be associated with mechanisms of defence and adaptation to the pathology [9]. miR-124 elevation was also observed in neuron-derived sEVs, and modulation was sensed in the cell secretome, recapitulating the intracellular levels. Interestingly, we also demonstrated that miR-124 was carried in sEVs from the donor neurons into the recipient microglial cells, leading to their reshaping and plasticity, with increased or decreased activation caused by the inhibitor and mimic, respectively [10]. In contrast with these data, our previous studies in mSOD1 MNs evidenced that when the miR-124 was restored to normal levels by transfecting ALS MNs with the miR-124 inhibitor, the secretome became neuroprotective [13]. Therefore, the beneficial or detrimental roles of miR-124 in AD and ALS, respectively, might be related to the type of neurons (neurons vs. MNs), the tested disease models, and the region of the central nervous system affected by the disorder.

Here, we tested the preconditioned secretome from anti-miR-124-treated mSOD1 MNs for its ability to halt motor disabilities, MN demise, neuron/glia deregulation, and muscle atrophy in mSOD1 mice. To achieve this, we performed an intrathecal injection of the preconditioned secretome in the lumbar section of the SC of mSOD1 mice at the early symptomatic stage (12-week-old) [70], and compared data with matched mSOD1 and WT controls injected with the vehicle (differentiated NSC-34 cell media with 1% FBS depleted in sEVs), as performed by others [25,26]. Such controls allowed us to evaluate the preventive effects of the preconditioned secretome injection on the pathological findings of mSOD1 mice at the symptomatic stage and to compare them with the normal behavioural and molecular status of the WT mice. By only injecting the vehicle, we were sure to not alter the course of the disease in the transgenic mice and to not cause any alteration in the WT animals, besides the eventual effect caused by the injection.

Cell secretomes are rich in soluble proteins and sEVs [71], which reach surrounding and distant cells and are engulfed by them [72]. sEVs and secretomes contain miRNAs [9,10,11,73] and therapeutic miRNA-enriched/depleted sEVs have emerged as novel therapeutics [17,74,75,76]. When administered in mice, the presence of sEVs/exosomes in several organs, including in the SC, was observed several hours after injection, until 72 h and disappearing at day 12 [25,30,77,78]. Since miR-124 elevation was observed in sEVs isolated from the secretome of mSOD1 MNs [11] and the sEVs also reflected the miR-124 modulation [13], we were interested in determining how long the labelled sEVs could be visualized in the SC, as we also had in mind their possible use in ALS therapeutics. Indeed, studies involving the intrathecal injection of PKH26-labelled sEVs from stem cells of different origins also identified their local distribution in the SC and their benefits in promoting recovery from spinal cord injuries and the inhibition of inflammation [25,79]. We observed that sEVs injected into the intrathecal space and labelled with the green probe PKH67 were visible in the SC at 8 and 72 h after administration, indicating that their dissemination and interaction with neural cells lasted for at least 3 days. This finding was important for inferring the benefits and lasting consequences of our preconditioned secretome administered by intrathecal injection in early symptomatic mSOD1 mice.

In previous studies, we found upregulated miR-124 in the SC of mSOD1 mice [36], which was also confirmed in the present study. Therefore, the next step was to validate that the preconditioned secretome from anti-miR-124-treated mSOD1 MNs was effective in regulating the expression of miR-124 toward normal values in the SC. In this study, we showed that the preconditioned secretome re-established the levels of miR-124 exhibited by the 15-week-old WT mice. The data were in conformity with our previous results demonstrating that such a secretome regulated the miR-124 levels in SC organotypic slices of mSOD1 mice [13]. This finding was important for us to proceed in our experimental design toward the establishment of the preconditioned secretome as a novel and promising therapeutic strategy for ALS.

Behavioural tests were performed on 14-week-old animals, and molecular assessments of the SC were carried out after 15-week-old mouse sacrifice, which preceded the advanced symptomatic stage that occurs at 16 weeks of age [70]. No approach for the modulation of miR-124 in SOD1-G93A transgenic mice, the best characterized model [80], has been performed before. The mice were phenotypically healthy until 80–90 days old, and then hindlimb tremors and weakness began, progressing to hyperreflexia and paralysis and culminating with death at around 120 days old [37,81]. Our results demonstrated that 14-week-old mSOD1 mice had several motor disabilities, including locomotor deficits that caused shorter steps, as previously demonstrated by other authors [38,39,40]. They also evidenced a lack of muscular strength, spending less time attached to the grid, as reported before [39,42,43,44]. Most of the ALS mice also evidenced an increase in spasticity by the retraction of the hindlimbs towards the abdomen and the closing of the fingers in a grasp, corresponding to the clasping and grasping reflexes, respectively. These disabilities associated with the corticospinal function might have been related to cerebellum dysfunction [82,83] and were observed previously in mSOD1 [41] and TDP-43 mouse models [84,85]. A less commonly used method, the cylinder test allowed us to evaluate the spontaneous activity and sensorimotor function of the mice, as described in [32,86]. We found significantly lower activity in the ALS mice compared to the WT mice, as demonstrated by the reduction in the number of times they reared up [44]. The injection of the preconditioned secretome prevented such motor disabilities from occurring, highlighting the importance of regulating miR-124 levels to preserve not only the MN function, but also the cell homeostasis in the SC [13] and, additionally, to maintain good coordination, balance, and motor functionality in mSOD1 mice.

The motor symptoms observed in ALS mice were unequivocally associated with muscular atrophy, the denervation of the neuromuscular junction, and the degeneration of the MNs. We observed a reduction in the area of gastrocnemius muscle fibers, together with MN loss and a lack of synaptic activity. Such a loss of muscle integrity is usually a result of MN demise [37,87]. However, other evidence points to defects in the muscle fibers that impair contractile function [46] and trigger MN degeneration [88]. These authors demonstrated severe skeletal muscle atrophy and the initiation of neuromuscular junction detachment from the muscle fibers caused by the expression of mSOD1 within skeletal muscle fibers. Moreover, Wong and Martin [88] showed that these mSOD1 mice developed weakness and abnormalities in motor function after performing hanging wire and clasping/grasping tests, corroborating our results. The loss of synaptic activity that we observed in the muscle contributed to the disassembly of the neuromuscular synapse, as previously demonstrated in ALS [89,90]. Again, the injection of the preconditioned secretome counteracted such alterations in the muscle and reinforced the significance of maintaining homeostatic miR-124 levels to support synaptic plasticity [91], as well as the myogenesis process, since the overexpression of miR-124 was shown to suppress myogenic differentiation [92].

We observed high levels of neurodegeneration and MN loss in the lumbar SC of symptomatic ALS mice, as previously published [36]. To contribute for these pathological features, the reduced expression of PSD-95 marked the disruption of postsynaptic signalling, as already described in mSOD1 mice [93] and evidenced in post mortem samples of ALS patients [94]. Thus, the retrograde and anterograde axonal transport systems seemed to be affected in the mSOD1 SC, with the decreased expression of dynein and kinesin, respectively. Such defects have been noticed in ALS mouse models and patients, as reviewed in [95]. Furthermore, we and others identified in the mSOD1 mouse model a reduction in kinesin-1 and dynein, together with significant spinal MN loss, reduced myelin fibers, and muscle pathology [96]. The axis between the neuronal fractalkine (CX3CL1) and its microglial receptor CX3CR1 is known to induce the production of soluble factors implicated in neuronal survival and microglia phagocytosis, thus exerting neuroprotective and immunoregulatory effects in mSOD1 mice [97]. Our results showed increased CX3CR1 but downregulated CX3CL1, as already described in [13,36], suggesting compromised CX3CL1/CX3CR1 neuronal–microglial communication that contributes to the progression of MN degeneration and pathophysiology in ALS [98].

Dysfunctional oligodendrocytes and myelin structures have been reported in the SC of ALS mouse models and patients, as reviewed in [99]. In accordance with published data [100], the reduced expression of MBP that we found in the SC of the ALS mice may have been implicated in the dysfunction of newly differentiated oligodendrocytes. Upregulated GPR17 levels additionally impact oligodendrocyte precursor cell (OPC) maturation due to its regulatory role [48]. Though important for the transition from OPCs to immature oligodendrocytes, GPR17 downregulation is mandatory to allow the maturation of OPCs, a process that is impaired in ALS [48]. Studies have demonstrated that toxic aggregates of SOD1 induce the demyelination of oligodendrocytes [101,102]. To compensate, OPCs increase their proliferation rate and differentiate into new mature oligodendrocytes. From this point of view, the upregulation of GPR17 is important for OPC maturation. However, the abnormal upregulation of this receptor in the SC of mSOD1 mice at the symptomatic stage precludes OPC terminal maturation [48]. For this reason, we suggest that upregulated GRP17 restrains oligodendrocyte maturation, which is further supported by the decreased gene levels of MBP and PLP that were observed. The preconditioned secretome from anti-miR-124-treated mSOD1 MNs prevented neurodegeneration occurring from weeks 12 to 15 in the mSOD1 mice, very likely by sustaining MN viability through the inhibition of early apoptosis [13]. Moreover, it also favoured postsynaptic signalling and myelin production, showing broad neuroprotection. The restoration of homeostatic miR-124 levels is crucial for optimal synaptic plasticity [91] and the prevention of demyelination [103]. Indeed, increased levels of miR-124 were shown to be associated with hippocampal demyelination and memory dysfunction, reinforcing the notion that its regulation is critical for the control of myelin processes [103].

Microglia activation and astrocyte reactivity contribute to the dissemination of neurodegeneration [49,50]. However, due to the glial cell heterogenous phenotypes, it has been hard to define the roles of microglia and astrocytes in pathological processes. Thus, our previous studies showed that several drivers of inflammation are upregulated in the SC of mSOD1 mice at the symptomatic stage [36] and that there is an immune imbalance in the SCOCs of such mice [13,36]. The slightly decreased expression of MFG-E8, TREM2, and P2RY12 indicated the decreased phagocytic ability of the microglial cells. Reduced MFG-E8 was identified in this model at both the pre-symptomatic and symptomatic stages [36], and TREM2 was suggested to be important in ALS, but this has been scarcely explored [3]. Moreover, downregulated P2RY12 has also been associated with the loss of microglia’s phagocytic abilities and cell migration [20]. Such results could be related to the upregulated miR-155 that we observed, considering the interference with microglia phagocytosis [20]. It should be noted that the preconditioned secretome tends to counteract the fingerprint of these parameters, indicating that they are characteristic of the symptomatic stage of mSOD1 mice and can be prevented. Such an effect may be related to the induced reduction we found for miR-155 together with the upregulation of arginase 1, supporting the existence of a more functional microglia after the intrathecal injection of the preconditioned secretome. Reduced levels of arginase 1, as we identified in the mSOD1 mice, have been shown to be linked to the existence of neuropathological mechanisms, neuroinflammation, and behavioural deficits in an AD mouse model [51] and to dysfunctional microglia in the SC of mSOD1 rats at disease end-stage [104]. The increased expression of Iba-1 microglia in mSOD1 mice is also associated with an activated microglia, and was found in our previous study [36]. Once more, the preconditioned secretome was effective in preventing such alterations from occurring in the mSOD1 mice at 15 weeks of age, thus supporting a more functional and reparative phenotype of microglia. Lately, the RNAseq analysis of microglia isolated from the SC of mSOD1 mice showed the existence of several activated phenotypes. A distinctive transcriptional signature, known as DAM, including the upregulation of TREM2 and TIMP2 and the downregulation of P2RY12, CX3CR1, and TMEM119, was identified [3,105,106]. The data we obtained for the dysregulation of P2RY12, CX3CR1, TMEM119, and TIMP2 gene expression levels suggested the existence of activated microglia and neuroinflammation [55,56,57], though we cannot discard the possibility that if evaluated for subtypes we could also identify the DAM microglia. However, with so many descriptions of microglia phenotypes in different diseases and through single-cell arrays, it was critical for us to investigate microglia functions instead of subtypes based on the gene signature. In summary, we observed activated microglia in the mSOD1 mice at the symptomatic stage with possible impairments in their phagocytic and migration abilities, which we believe to favour neurodegeneration. Remarkably, the preconditioned secretome was effective in preventing such pathological alterations in the microglia. When injected into 12-week-old pre-symptomatic mice, the secretome was shown to delay disease progression and sustain most of the markers associated with the neural cell steady state in 15-week-old animals.

We observed in a previous study that symptomatic mSOD1 mice also exhibited upregulated levels of miR-155 and miR-146a, alongside miR-124, in the SC, which are known to contribute to neuroinflammation [36]. Furthermore, another miRNA that was shown to regulate microglia activation and MN death in ALS was miR-125b [107]. Likewise, miR-21 was found to be upregulated in mSOD1 mice [108]. All of these, except for miR-125b, were upregulated in the SC of the symptomatic mSOD1 mice. Again, the preconditioned secretome prevented such dysregulation from occurring, clearly highlighting the anti-inflammatory efficacy of its intrathecal administration in the ALS mouse model. Ongoing studies are being performed with tricultures of spinal microglia, astrocytes, and MNs in microfluidic devices to better explore the signalling mechanisms and gain of function by the preconditioned secretome and miRNA-loaded sEVs, with relevant application to the development of effective target-driven therapies [109,110]. Our results also supported increased astrocyte reactivity and potential neurotoxic effects in the SC of mSOD1 mice, based on the increased levels of GFAP, Cx43, S100B, and vimentin, in agreement with other studies [35,36]. The preconditioned secretome completely averted the appearance of such a pathological astrocyte phenotype, highlighting its immune remodelling ability, if we also consider its preventive effects against TNF-α upregulation and IL-10 downregulation, which are known as inflammatory markers [13,36].

## 5. Conclusions

Overall, this study validated the therapeutic potential of the secretome derived from anti-miR-124-treated mSOD1 MNs in the prevention of pathological neural cell signatures in symptomatic mSOD1 mice and the subsequent dysregulation of the paracrine signalling implicated in disease dissemination. From a single intrathecal injection of the preconditioned secretome carrying basal levels of miR-124 at the early symptomatic stage of the mSOD1 mice, we were able to regulate miR-124 toward normal values. With such an approach, we sustained anti-inflammatory/phagocytic microglia, MN function, myelination, astrocyte neuroprotection, and immune regulation at 3 weeks after injection, i.e., at the symptomatic stage wherein all these mechanisms turn pathological. Importantly, the secretome also preserved synaptic signalling, functional skeletal muscle, and neuromuscular junction, which translated to the preservation of motor coordination, balance, corticospinal function, and spontaneous behaviour in the symptomatic mSOD1 mice. Our preconditioned secretome based on the strict regulation of miR-124 levels showed immunoregulatory and potential neuroregenerative properties, opening a new avenue for developing a novel effective therapeutic strategy that can be translated into clinical applications in the future. To achieve this, it will be crucial to stratify patients with upregulated levels of miR-124 based on MNs differentiated from induced pluripotent stem cells (iPSCs) or neural precursor cells generated from their somatic cells. In this sense, we propose the modulation of upregulated miR-124 expression levels in such MNs to produce the preconditioned personalized secretome. We can envisage, as was lately described for the secretome of iPSCs [19], that our preconditioned secretome might be used as an autologous treatment in judiciously selected patients to halt ALS pathogenicity or at least delay disease progression.

## Figures and Tables

**Figure 1 biomedicines-10-02120-f001:**
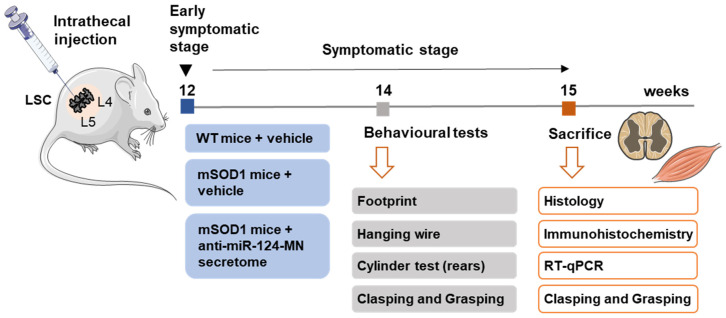
Experimental design. The wild-type (WT) and SOD1-G93A (mSOD1) mice were injected in the lumbar spinal cord [31] at the early symptomatic stage (12-week-old), either with the MN medium (vehicle, control group) or with the secretome from anti-miR-124-treated mSOD1 MNs (only the mSOD1 mice). Two weeks later, animals were behaviourally characterized through footprint, hanging wire, cylinder, clasping, and grasping tests. At 15 weeks of age, the animals were sacrificed, and the lumbar spinal cord [31] and gastrocnemius muscle were isolated for histological and immunohistological analysis, as well as for reverse transcription quantitative real-time polymerase chain reaction (RT-qPCR) and western blot evaluations. This Figure was partially created with Servier Medical Art (smart.servier.com).

**Figure 2 biomedicines-10-02120-f002:**
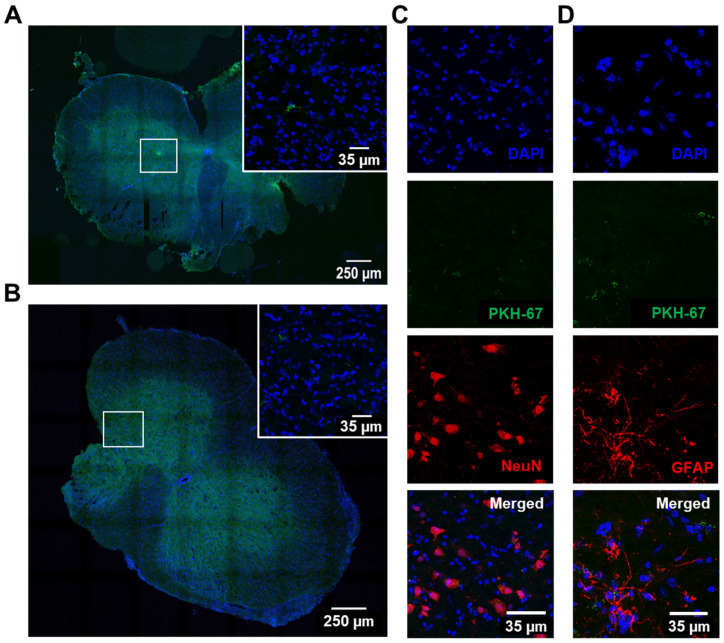
Intrathecal injection of labelled small extracellular vesicles (sEVs) in 12-week-old wild-type (WT) animals leads to their dissemination and interaction with nerve cells, as observed in the lumbar spinal cord (SC) sections at 8 h and 72 h after administration. sEVs were isolated by differential ultracentrifugation and labelled with PKH67 cell linker before injection in the WT mice, which were sacrificed at 8 h and 72 h thereafter. (**A**,**B**) Representative images of transversal SC slices with labelled sEVs (green) and respective insets at (**A**) 8 h and (**B**) 72 h post-injection. (**C**,**D**) Representative images of PKH67-labelled sEVs distributed among (**C**) NeuN-stained cells and (**D**) GFAP-stained cells from the lumbar SC of WT mice 72 h after injection. The nuclei were stained with DAPI (blue). DAPI, 4′,6-diamidino-2-phenylindole; GFAP, glial fibrillary acidic protein; NeuN, hexaribonucleotide binding protein 3.

**Figure 3 biomedicines-10-02120-f003:**
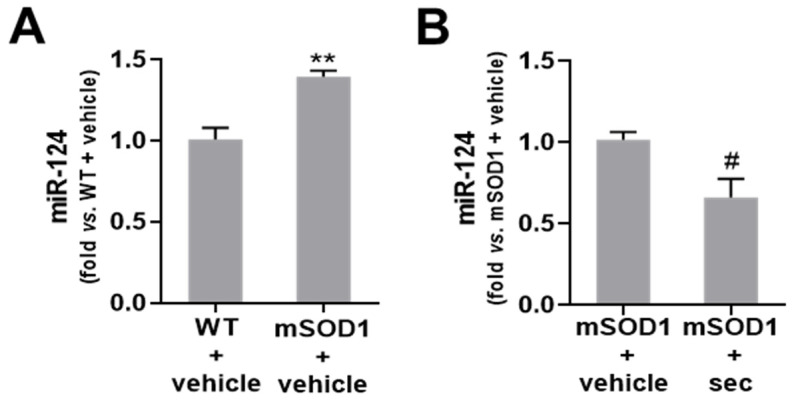
The secretome from mSOD1 MNs transfected with anti-miR-124 abolishes the upregulation of miR-124 in the spinal cord (SC) of ALS mice after 3 weeks of intrathecal injection. Expression of miRNA-124 in the lumbar SC of (**A**) SOD1-G93A (mSOD1) mice injected at 12 weeks of age with the vehicle (basal media of NSC-34 motor neurons (MNs)) in comparison with the respective wild-type (WT) animals, and (**B**) mSOD1 mice injected with the secretome derived from anti-miR-124-treated mSOD1 MNs (mSOD1 + sec) in comparison with those injected only with the vehicle. The results were obtained at 15 weeks of age and were normalized to SNORD110. Data are expressed as fold change vs. (**A**) WT + vehicle and (**B**) mSOD1 + vehicle (mean ± SEM) from at least 5 animals per group. ** *p* < 0.01 vs. WT + vehicle; # *p* < 0.05 vs. mSOD1 + vehicle, unpaired and parametric *t*-test (with Welch’s correction when needed).

**Figure 4 biomedicines-10-02120-f004:**
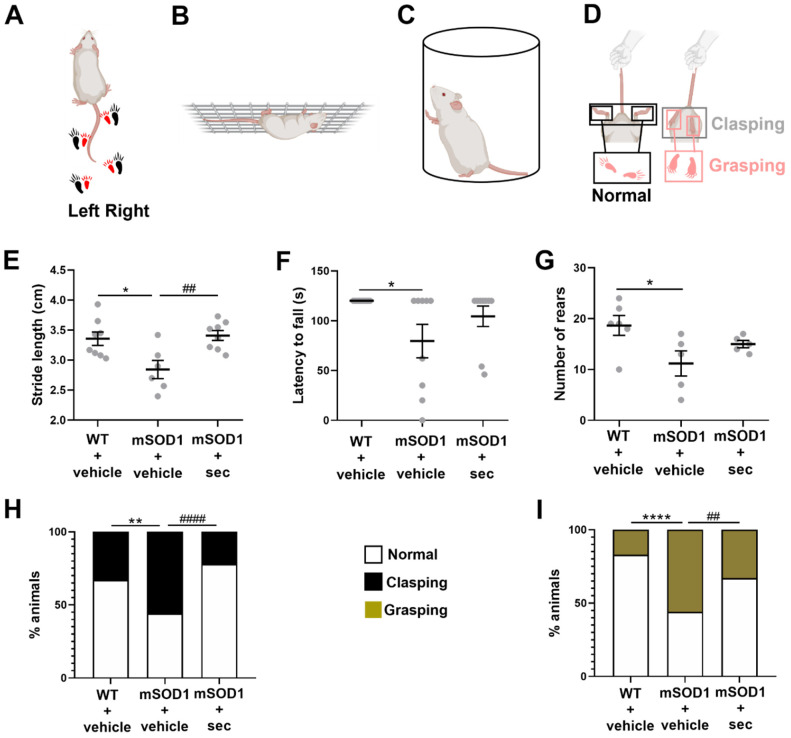
Motor performance, muscular strength, spontaneous activity, and corticospinal function in ALS mice are improved after 2 weeks of intrathecal injection of the secretome from anti-miR-124-treated mSOD1 motor neurons (MNs). Representative illustrations of (**A**) footprint test, (**B**) hanging wire test, (**C**) cylinder test, and (**D**) clasping/grasping reflexes test. Measurement of the (**E**) stride length (in centimeters, cm), (**F**) time holding onto the cage grid (in seconds, s), (**G**) number of times the mice reared up against the cylinder, and percentage (%) of animals showing the (**H**) clasping and (**I**) grasping reflexes in wild-type (WT)/SOD1-G93A (mSOD1) mice injected with the vehicle (basal media of NSC-34 MNs) and mSOD1 mice injected with the secretome derived from mSOD1 MNs modulated with anti-miR-124 (mSOD1 + sec). Data are expressed as mean ± SEM for (**E**–**G**) and percentage (%) for (**H**,**I**) from at least 5 animals per group. **** *p* < 0.0001, ** *p* < 0.01 and * *p* < 0.05 vs. WT + vehicle; #### *p* < 0.0001 and ## *p* < 0.01 vs. mSOD1 + vehicle. One-way ANOVA followed by multiple-comparisons Bonferroni post hoc correction was used for footprint and cylinder tests; unpaired and one-way non-parametric ANOVA (Kruskal–Wallis test) was used for hanging wire test; chi-square (and Fisher’s exact) test was used for clasping and grasping tests. Panels (**A**–**D**) were partially created with Biorender (Biorender.com).

**Figure 5 biomedicines-10-02120-f005:**
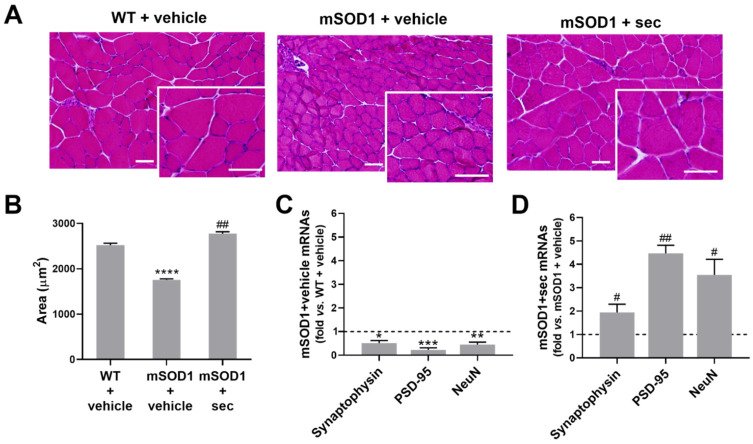
Intrathecal injection of anti-miR-124-treated ALS MN-derived secretome in 12-week-old mSOD1 mice prevents loss of muscle fiber area and the deregulation of genes that direct synaptic proteins at 3 weeks after treatment. (**A**) Representative images of transversal sections of gastrocnemius muscle from the wild-type (WT) and SOD1-G93A (mSOD1) mice injected with the vehicle (basal media of NSC-34 motor neurons (MNs)), as well as mSOD1 mice injected with the secretome derived from anti-miR-124-treated mSOD1 MNs (mSOD1 + sec), stained with hematoxylin–eosin. Scale bars: 50 µm. (**B**) Respective average muscle fiber area (in µm). (**C**) Gene expression of synaptophysin, PSD-95, and NeuN from mSOD1 mice injected with the vehicle in comparison with the respective WT mice, and (**D**) from mSOD1 mice injected with the secretome in comparison with those injected with the vehicle. Results are mean (±SEM) for (**B**) and expressed as fold change vs. WT + vehicle for (**C**) or fold change vs. mSOD1 + vehicle for (**D**). The images from three fields of the muscle per animal (from three animals per group) were used for histological analysis and five animals per group for RT-qPCR analysis. **** *p* < 0.0001, *** *p* < 0.001, ** *p* < 0.01, and * *p* < 0.05 vs. WT + vehicle; ## *p* < 0.01 and # *p* < 0.05 vs. mSOD1 + vehicle. One-way ANOVA followed by multiple-comparisons Bonferroni post hoc correction was used for (**B**) and unpaired and parametric *t*-test with Welch’s correction for (**C**,**D**). NeuN, hexaribonucleotide binding protein 3; PSD-95, postsynaptic density protein 95.

**Figure 6 biomedicines-10-02120-f006:**
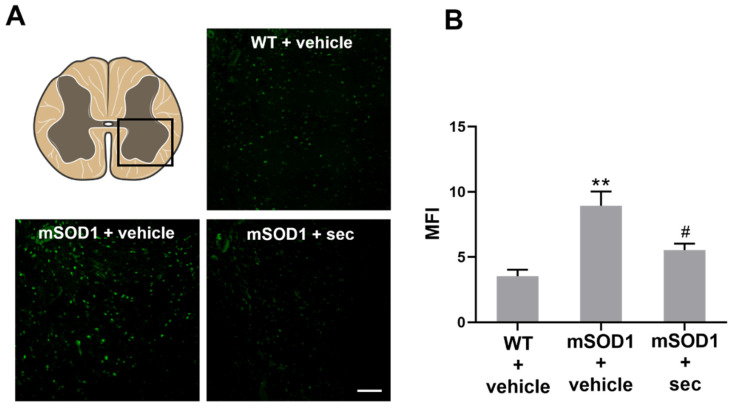
Intrathecal injection of the secretome from anti-miR-124-modulated mSOD1 motor neurons (MNs) in the lumbar spinal cord of mSOD1 mice at 12 weeks of age prevents age-associated neurodegeneration after 3 weeks of its administration. (**A**) Representative images of the ventral horn of the lumbar section grey matter stained by Fluoro-Jade B fluorescence (square) from 15-week-old wild-type (WT)/SOD1-G93A (mSOD1) mice injected with the vehicle (basal media of NSC-34 MNs) and mSOD1 mice injected with the secretome derived from mSOD1 MNs modulated with anti-miR-124 (mSOD1 + sec); (**B**) the respective quantification of mean fluorescence. Scale bar: 100 µm. Data from 3 animals per group are expressed as fold change (mean ± SEM) vs. WT + vehicle. ** *p* < 0.01 vs. WT + vehicle; # *p* < 0.05 vs. mSOD1 + vehicle, one-way ANOVA followed by multiple-comparisons Bonferroni post hoc correction. MFI, mean fluorescence intensity.

**Figure 7 biomedicines-10-02120-f007:**
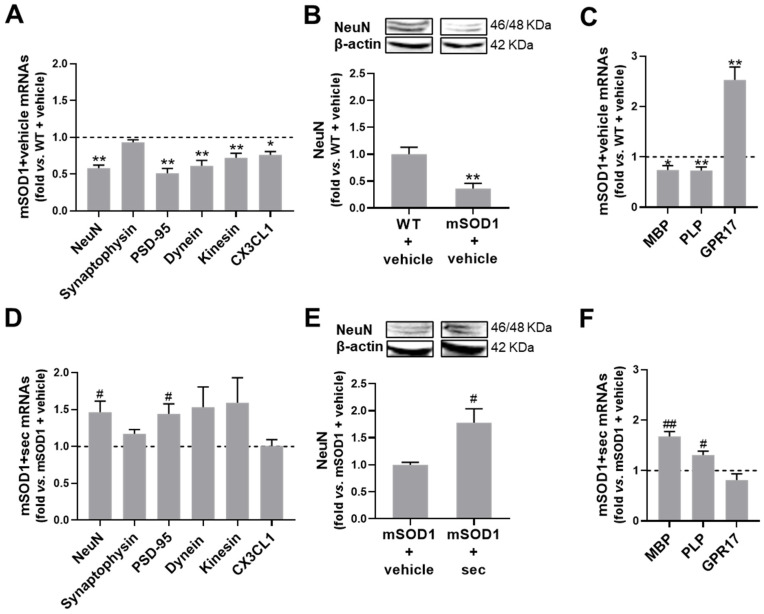
Neuronal demise and deficits in synaptic signalling, axonal transport, CX3CL1-CX3CR1 axis, and myelination in the lumbar spinal cord (SC) of 15-week-old mSOD1 mice are prevented by the intrathecal injection of the secretome from modulated mSOD1 motor neurons (MNs) in 12-week-old animals. (**A**) Gene expression of neuronal-related NeuN, synaptophysin, PSD-95, dynein, kinesin, and CX3CL1; (**B**) protein expression of NeuN; and (**C**) myelin-associated genes (MBP, PLP, and GPR17) in the SC of 15-week-old SOD1-G93A (mSOD1) mice that were treated with the vehicle (basal media of NSC-34 MNs) at 12-week-old, in comparison with the respective wild-type (WT) values. (**D**–**F**) Data from matched experiments realized in mSOD1 mice injected with the secretome derived from anti-miR-124-treated mSOD1 MNs (mSOD1 + sec) in comparison to those treated with the vehicle. (**B**,**E**) Representative results from one blot. The results were normalized to RPL-19 for RT-qPCR and β-actin for western blot. Data are expressed as fold change vs. (**A**–**C**) WT + vehicle and (**D**–**F**) mSOD1 + vehicle (mean ± SEM) from at least 5 animals per group. ** *p* < 0.01 and * *p* < 0.05 vs. WT + vehicle; ## *p* < 0.01 and # *p* < 0.05 vs. mSOD1 + vehicle, unpaired and parametric *t*-test (with Welch’s correction when needed). CX3CL1, C-X3-C motif chemokine ligand 1/fractalkine; GPR17, G-protein-coupled receptor 17; MBP, myelin basic protein; NeuN, hexaribonucleotide-binding protein 3; PLP, myelin proteolipid protein; PSD-95, postsynaptic density protein 95; RPL19, 60S ribosomal L19; RT-qPCR, reverse transcription quantitative real-time polymerase chain reaction.

**Figure 8 biomedicines-10-02120-f008:**
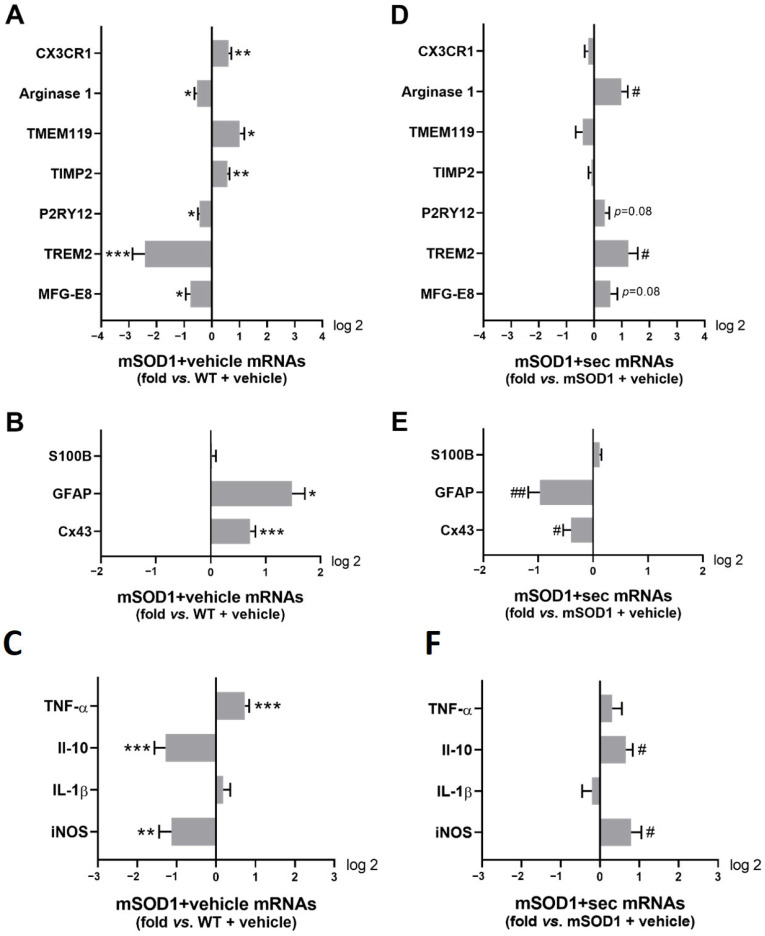
Intrathecal injection of the secretome from anti-miR-124-modulated mSOD1 motor neurons (MNs) in the lumbar spinal cord (SC) of mSOD1 mice at 12-week-old prevents microglia activation, astrocyte reactivity, TNF-α signalling, and inflammation associated with the symptomatic stage. Gene expression of (**A**) microglia-associated markers (MFG-E8, TREM 2, P2RY12, TIMP2, arginase 1, and CX3CR1); (**B**) astrocyte-related markers (Cx43, GFAP, and S100B); and (**C**) inflammatory-associated markers (iNOS, IL-1β, IL-10, and TNF-α) in the SC of SOD1-G93A (mSOD1) mice injected with the vehicle (basal media of NSC-34 MNs) in comparison to those in wild-type (WT) animals; (**D**–**F**) after the administration of the secretome derived from anti-miR-124-treated mSOD1 MNs (mSOD1 + sec) as compared with mSOD1 mice treated with the vehicle. The results were normalized to RPL-19. Data are expressed as fold change vs. (**A**–**C**) WT + vehicle and (**D**–**F**) mSOD1 + vehicle (mean ± SEM) from at least 5 animals per group. *** *p* < 0.001, ** *p* < 0.01, and * *p* < 0.05 vs. WT + vehicle; ## *p* < 0.01 and # *p* < 0.05 vs. mSOD1 + vehicle, unpaired and parametric *t*-test (with Welch’s correction when needed). CX3CR1, c-x3-c chemokine receptor 1; Cx43, connexin 43; GFAP, glial fibrillary acidic protein; IL-10, interleukin 10; IL-1β, interleukin 1β; iNOS, inducible nitric oxide synthase; MFG-E8, milk fat globule epidermal growth factor 8; P2RY12, purinergic receptor p2y12; RPL19, 60S ribosomal L19; S100B, S100 calcium-binding protein B; TIMP2, tissue inhibitor of metalloproteinases 2; TMEM119, transmembrane protein 119; TNF-α, tumour necrosis factor alpha; TREM2, triggering receptor expressed on myeloid cells 2.

**Figure 9 biomedicines-10-02120-f009:**
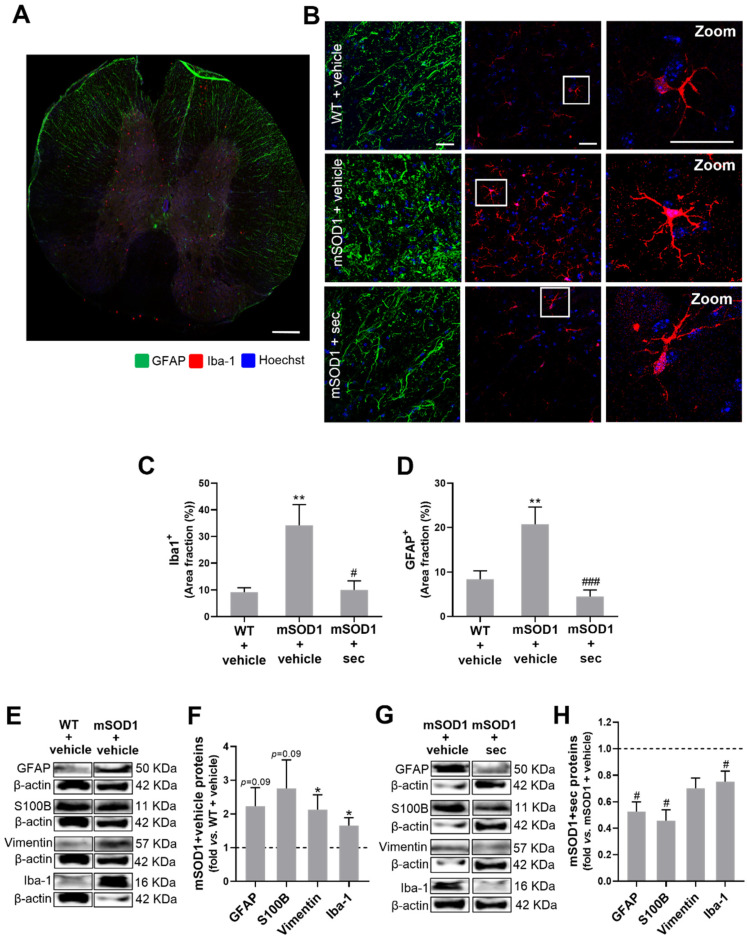
Intrathecal injection of the secretome from anti-miR-124-modulated mSOD1 motor neurons (MNs) in the lumbar spinal cord (SC) of mSOD1 mice at 12-week-old counteracts the upregulation of proteins linked to glia-driven immunoreactivity processes after 3 weeks of its administration. (**A**) Representative image of a transversal SC slice stained with Iba-1 (microglia-associated marker) and glial fibrillary acidic protein/GFAP (astrocyte-associated marker). (**B**) Representative images for GFAP- (green) and Iba-1- (red) positive cells from the lumbar SC of WT and SOD1-G93A (mSOD1) mice injected with the vehicle (basal media of NSC-34 MNs) and mSOD1 mice injected with the secretome derived from anti-miR-124-treated mSOD1 MNs (mSOD1 + sec). A zoomed-in image of an Iba-1-positive cell is shown. Nuclei were stained with Hoechst (blue). Scale bar: (**A**) 300 µm and (**B**) 30 µm. (**C**,**D**) Area fraction (in percentage, %) occupied by GFAP- and Iba-1-positive cells, respectively. (**E**,**G**) Representative western blots (WB). (**F**,**H**) Data resulting from WB analysis of reactive astrocytic markers (GFAP, S100B, and vimentin) and microglial Iba-1. β-actin was used as a loading control for WB analysis. Results are mean (± SEM) for (**C**,**D**) and expressed as fold change vs. WT + vehicle for (**F**) or fold change vs. mSOD1 + vehicle for (**H**). The images were analysed from five ventral horn fields per animal (from three animals per group) for immunohistochemistry and five animals per group for WB analysis. ** *p* < 0.01 and * *p* < 0.05 vs. WT + vehicle; ### *p* < 0.001, and # *p* < 0.05 vs. mSOD1 + vehicle. One-way ANOVA followed by multiple-comparisons Bonferroni post hoc correction was used for (**C**,**D**), and unpaired and parametric *t*-test with Welch’s correction for (**F**,**H**). GFAP, glial fibrillary acidic protein; Iba-1, ionized calcium-binding adaptor molecule 1; S100B, S100 calcium-binding protein B.

**Figure 10 biomedicines-10-02120-f010:**
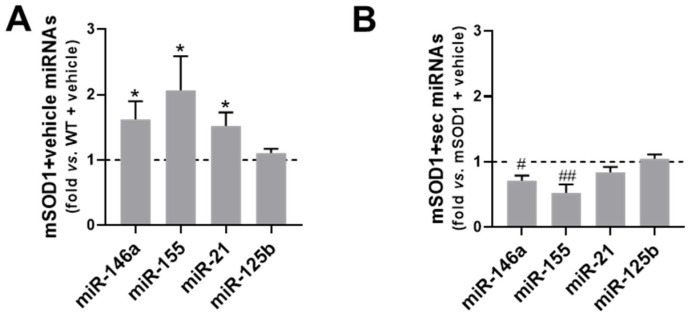
Upregulation of inflammatory-associated miRNA observed in the lumbar spinal cord (SC) of ALS mice is prevented by the secretome from the mSOD1 MNs engineered with anti-miR-124. Expression of inflammatory-associated micro(mi)RNAs (miR-146, miR-155, miR-21, miR-124, and miR-125b) in the SC of (**A**) SOD1-G93A (mSOD1) mice injected with the vehicle (basal media of NSC-34 motor neurons (MNs)) in comparison with those in wild-type (WT) animals, and (**B**) mSOD1 mice injected with the secretome derived from anti-miR-124-treated mSOD1 MNs (mSOD1 + sec) in comparison with those treated with the vehicle. The results were normalized to SNORD110. Data are expressed as fold change vs. (**A**) WT + vehicle and (**B**) mSOD1 + vehicle (mean ± SEM) from at least 5 animals per group. * *p* < 0.05 vs. WT + vehicle; ## *p* < 0.01 and # *p* < 0.05 vs. mSOD1 + vehicle, unpaired and parametric *t*-test (with Welch’s correction when needed).

## Data Availability

Data is contained within the article.

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
