# Peer review of "Intrathecal Injection of the Secretome from ALS Motor Neurons Regulated for miR-124 Expression Prevents Disease Outcomes in SOD1-G93A Mice"

_biomedicines, 2022, doi:10.3390/biomedicines10092120_

Round 1

Reviewer 1 Report

The manuscript by Barbosa et al. shows interesting data on the possible role of the secretome of anti-miR-124 transfected mSOD1 motor neurons in the prevention of disability in an amyotrophic lateral sclerosis model consisting of a mouse with a point mutation from glycine to alanine in the superoxide dismutase 1 gene.

The main problem in this manuscript which is a continuation of other publications of the group is that in the experimental design there is no WT + anti-miR-124-MN secretome group. To draw a strong conclusion one needs to have all the information and in this case it is not possible.

Another important issue is that the results section is too long, there is too much information: introduction+data+discussion in each sub-section. Shorten the results with just the data and maybe one line with some kind of introduction.

Other problems:

1) Line 59, it is stated that miRNAs can be released into the secretome as free species or encapsulated in small extracellular vesicles. A reference is needed because references 8 and 9 refer to other work by the authors and not to what is being said in particular.

2) Line 163. Explain why WT animals injected with sEVs derived from WT MNs were sacrificed 8h and 72h after injection.

3) Line 241. ...followed by 50 cycles of amplification...If 50 cycles are needed there may be a problem with detection. Check that the Ct is not higher than 30.

4) Line 278...The cell nucleus was stained with Hoechst...Are you sure you have used Hoechst or DAPI because in the figures it is DAPI.

5) Line 336 ...were internalised by the neurons while staying in the proximity of the astrocytes...It is impossible to see this in the images in figure 2. It is very important to clarify this because the interpretation of this parameter can change the discussion and conclusions.

6) Figures 3, 4, 5, 6, 7: The mSOD1+ vehicle group is shown as a condition in one panel and as a control normalised to 1 in the other panel. The data shown in these graphs should be shown with all 3 groups together and statistical analysis performed should be using one-way ANOVA and Bonferroni post hoc. This is also very important for interpreting the data and may change the conclusions.

7) Typo: Line 594 iba-1 is Fig 9.

Author Response

We thank the Reviewer for the insightful comments and suggestions to improve the manuscript. We have addressed all the criticisms in the manuscript (marked with “Track Changes” in the revised version, as it follows:

The main problem in this manuscript which is a continuation of other publications of the group is that in the experimental design there is no WT + anti-miR-124-MN secretome group. To draw a strong conclusion one needs to have all the information and in this case it is not possible.

We understand the reviewer concerns, but our decision was based on the fact that the WT samples do not behave as the pathological ones and do not have increased miR-124 that justifies the use of an inhibitor for miR-124.

Here, we intend to strictly regulate miR-124 toward the steady-state (homeostatic levels). This aim cannot be achieved in the WT because the expression of miR-124 is already regulated, so miR-124 downregulation by the anti-miR-124 will induce harmful effects and cannot be a clean control as the vehicle we used.

This is also the reason to not silencing completely the miR-124 in the pathological samples. Actually, we should always consider that this major neuronal miR-124 has important roles, because it is necessary to regulate many functions, such as neuronal differentiation/maturation and the regulation of synaptic activity [1], so the levels should be kept near the WT ones.

For that reason, reduction of miR-124 should only be undertaken when the levels are abnormally upregulated in motor neurons (MNs), what is the case of mSOD1 MNs and not WT MNs. Moreover, results from our lab showed that the transfection of neurons derived from iPSCs generated from an healthy control with miR-124 inhibitor compromised  the dendritic spine density [2]. Other studies also evidenced the consequences of the deprivation of miR-124 in the prefrontal cortex functionality [3] and in neural specification and differentiation [4].

Once the idea is to use autologous cell-free transplantation in patients stratified for having increased miR-124 in their fibroblast-derived-MNs, our objective was to show that the progression of the disease was delayed or prevented in ALS mice injected with the secretome having regulated levels of miR-124 relatively to the natural evolution of WT mice + vehicle and mSOD1 + vehicle, i.e. what was used in the culture of MNs treated with the anti-miR-124.

Indeed, in our prior in vitro studies, the use of anti-miR-124 in mSOD1 MNs, showed to prevent the dysregulation of neurite network, mitochondria dynamics and axonal transport in these cells. In addition we also obtained a secretome with regulated miR-124 levels close to the WT cells cultured with the vehicle [5]. The beneficial effects of such preconditioned secretome from anti-miR-124 mSOD1 MNs was similarly sensed in preventing ALS-associated pathological features in organotypic slices from the early symptomatic ALS mice, thus pointing to benefits in rescuing MN/glial dysfunction whenever miR-124 reveals to be upregulated.

Based on such previous data, we aimed here to validate such promising results in the in vivo ALS model.

To also point that the secretome from the WT MNs has normal neuronal growth factors that benefit MN survival and prevents some MN degeneration. This is the reason of some clinical trials that are using enriched media with NGF and BDNF in clinical trials for ALS. Therefore, again it was not reasonable to use such secretome as control as suggested by the other reviewers. Use of vehicle group as control for the injection of exosomes was also performed in similar studies [6,7].

We hope to have make clear that any other control we could use was not indicated to evidence that the preconditioned secretome from ALS MNs treated with anti-miR-124 prevents the progression of the natural disease observed with mSOD1 + vehicle until the symptomatic stage, which can only be compared with its respective matched control, i.e., WT + vehicle to be in similar conditions. Finally, we are not proposing to use such preconditioned secretome from anti-miR-124 ALS MNs in patients revealing normal levels of miR-124, only in those showing upregulated miR-124. It wouldn’t make sense considering the crucial roles of miR-124 in sustaining neural cell function, hence the reason not to use the control: WT + anti-miR-124-MN secretome group as proposed by the reviewer. We also added an explanation in the revised manuscript (lines 244-256 and 1054-1062).

Another important issue is that the results section is too long, there is too much information: introduction+data+discussion in each sub-section. Shorten the results with just the data and maybe one line with some kind of introduction.

We thank the Reviewer for calling our attention to such issue and in agreement with his/her suggestion we shortened the “Results” section and pertinent aspects associated to the Discussion were moved to such section.

Other problems:

1) Line 59, it is stated that miRNAs can be released into the secretome as free species or encapsulated in small extracellular vesicles. A reference is needed because references 8 and 9 refer to other work by the authors and not to what is being said in particular.

We understand the Reviewer’s concern and we included associated references (now References 8-11; previous ones are now references 11,12) showing the recapitulation of miRNA modulation in both the two fractions of the secretome, i.e., free species (sEV-free) and included in sEVs/exosomes (sEV fraction), demonstrating the different ways of miRNA release from cells, with close influence on the environmental cells (free species) and distant cells (sEVs).

2) Line 163. Explain why WT animals injected with sEVs derived from WT MNs were sacrificed 8h and 72h after injection.

Our previous results performed in vitro showed the ability of the sEVs derived from mSOD1 MNs to be engulfed by microglia cells, promoting different microglial phenotypes from 2 h to 4 h and 24 h of incubation [8]. In another study [9], we noticed that exosomes colocalize with microglia lysosomes at 24 h and that disappeared after 72 h, probably by autophagy (unpublished). These studies deal with the microglial cells that have a capacity to phagocytose and degrade foreign material. In other studies, however, bioluminescence of labeled exosomes/sEVs was time-dependently reduced but remained until 72h after intravenous injection in the whole-body [10]. Interestingly the free dye in such study disappeared after 24h. In tissue sections (spleen, kidney, and heart), collected 72 h after the animal’s sacrifice, the exosomes were observed but not the free dye. In another study the intrathecal injected sEVs from Wharton’s jelly stem cells lasted for one week in the spinal cord as detected by the size of labeled vesicles. Here, the control animals were also injected with the vehicle [6]. Additional studies using intravenous injection of exosomes for cancer therapy, showed the presence of vesicles in major organs (liver, spleen, lung, heart, and kidney) and tumors at 12, 24, 48, and 72 hours after injection, and again a vehicle injection was used as a control [7]. Also, injected sEVs through the tail vein for liver regeneration were detected at 1 h postinjection, peaked at 24 h and gradually decreased, but were clearly distinguished in the liver until day 6 and disappeared at day 12 [11]. Finally, to highlight that intranasally administered exosomes derived from marrow mesenchymal cells were found in the pathological murine brain (stroke, autism, Parkinson’s disease and Alzheimer’s disease) 96 h post administration [12].

In conformity we clarified/justified this issue in section 2.3 - lines 277-284 of the revised manuscript.

3) Line 241. ...followed by 50 cycles of amplification...If 50 cycles are needed there may be a problem with detection. Check that the Ct is not higher than 30. 

We apologize for our mistake. Our standard running conditions for RT-qPCR concerning the number of cycles is 40 for gene amplification and 50 for miRNA amplification, as usually performed in our lab [13,14]. However, our amplification Cts were not higher than 30 cycles for genes and 40 for miRNAs.

This is now corrected and clarified in the revised manuscript – section 2.6 – lines 388 and 395.

4) Line 278...The cell nucleus was stained with Hoechst...Are you sure you have used Hoechst or DAPI because in the figures it is DAPI.

The cell nucleus of neurons and astrocytes in Figure 2 were stained with DAPI, while the cell nucleus of the microglia and astrocytes from the Figure 9A,B,F were labelled with Hoechst. The type of staining of cell nucleus were clarified in the subsection 2.7 of the Methods (lines 440-442) and included in Figure captions of the revised manuscript.

5) Line 336 ...were internalised by the neurons while staying in the proximity of the astrocytes...It is impossible to see this in the images in figure 2. It is very important to clarify this because the interpretation of this parameter can change the discussion and conclusions.

We understand the Reviewer’s comment and we apologize for the wrong sentence. Such indication was removed, and the sentence was rephrased in the revised manuscript (lines 511-513).

6) Figures 3, 4, 5, 6, 7: The mSOD1+ vehicle group is shown as a condition in one panel and as a control normalised to 1 in the other panel. The data shown in these graphs should be shown with all 3 groups together and statistical analysis performed should be using one-way ANOVA and Bonferroni post hoc. This is also very important for interpreting the data and may change the conclusions.

We agree with the reviewer that the comparison of the three groups in one graph are helpful for interpreting the data. Therefore, we have modified Figures 4, 5, 6 and 9 accordingly, using the raw data obtained by the direct counting/measurement of the results for each condition, namely in the behavioral tests, the fiber area of the muscle, the Fluorojade B positive neurons and the GFAP/Iba1 positive cells from the immunohistochemistry analysis, as well as the respective legends and the Results section.

Concerning the results obtained from western blot and RT-qPCR assays, we kept the comparison of mSOD1 mice + vehicle vs. WT mice + vehicle and mSOD1 mice + sec vs. mSOD1 mice + vehicle, because both experiments consider the comparison relatively to a given control and our main goal was to compare the specific effects of the preconditioned secretome in preventing disease progression and  mSOD1 mice disabilities. However, to add further information for the interpretation of these results, we performed a one-way ANOVA and Bonferroni post hoc analysis with all 3 groups together, as now depicted in the new Supplementary Table S4.

7) Typo: Line 594 iba-1 is Fig 9.

We thank the Reviewer for identifying the mistake. Correct indication is now included in the revised manuscript (line 900).

  1. Sun, A.X.; Crabtree, G.R.; Yoo, A.S. MicroRNAs: regulators of neuronal fate. Curr Opin Cell Biol 2013, 25, 215-221, doi:10.1016/j.ceb.2012.12.007.
  2. Garcia, G.; Pinto, S.; Cunha, M.; Fernandes, A.; Koistinaho, J.; Brites, D. Neuronal Dynamics and miRNA Signaling Differ between SH-SY5Y APPSwe and PSEN1 Mutant iPSC-Derived AD Models upon Modulation with miR-124 Mimic and Inhibitor. Cells 2021, 10, doi:10.3390/cells10092424.
  3. Kozuka, T.; Omori, Y.; Watanabe, S.; Tarusawa, E.; Yamamoto, H.; Chaya, T.; Furuhashi, M.; Morita, M.; Sato, T.; Hirose, S.; et al. miR-124 dosage regulates prefrontal cortex function by dopaminergic modulation. Sci Rep 2019, 9, 3445, doi:10.1038/s41598-019-38910-2.
  4. Sun, K.; Westholm, J.O.; Tsurudome, K.; Hagen, J.W.; Lu, Y.; Kohwi, M.; Betel, D.; Gao, F.B.; Haghighi, A.P.; Doe, C.Q.; et al. Neurophysiological defects and neuronal gene deregulation in Drosophila mir-124 mutants. PLoS Genet 2012, 8, e1002515, doi:10.1371/journal.pgen.1002515.
  5. Vaz, A.R.; Vizinha, D.; Morais, H.; Colaço, A.R.; Loch-Neckel, G.; Barbosa, M.; Brites, D. Overexpression of miR-124 in Motor Neurons Plays a Key Role in ALS Pathological Processes. Int J Mol Sci 2021, 22, doi:10.3390/ijms22116128.
  6. Noori, L.; Arabzadeh, S.; Mohamadi, Y.; Mojaverrostami, S.; Mokhtari, T.; Akbari, M.; Hassanzadeh, G. Intrathecal administration of the extracellular vesicles derived from human Wharton's jelly stem cells inhibit inflammation and attenuate the activity of inflammasome complexes after spinal cord injury in rats. Neurosci Res 2021, 170, 87-98, doi:10.1016/j.neures.2020.07.011.
  7. Zhang, J.; Ji, C.; Zhang, H.; Shi, H.; Mao, F.; Qian, H.; Xu, W.; Wang, D.; Pan, J.; Fang, X.; et al. Engineered neutrophil-derived exosome-like vesicles for targeted cancer therapy. Sci Adv 2022, 8, eabj8207, doi:10.1126/sciadv.abj8207.
  8. Pinto, S.; Cunha, C.; Barbosa, M.; Vaz, A.R.; Brites, D. Exosomes from NSC-34 Cells Transfected with hSOD1-G93A Are Enriched in miR-124 and Drive Alterations in Microglia Phenotype. Front Neurosci 2017, 11, 273, doi:10.3389/fnins.2017.00273.
  9. Fernandes, A.; Ribeiro, A.R.; Monteiro, M.; Garcia, G.; Vaz, A.R.; Brites, D. Secretome from SH-SY5Y APPSwe cells trigger time-dependent CHME3 microglia activation phenotypes, ultimately leading to miR-21 exosome shuttling. Biochimie 2018, 155, 67-82, doi:10.1016/j.biochi.2018.05.015.
  10. Lee, B.R.; Kim, J.H.; Choi, E.S.; Cho, J.H.; Kim, E. Effect of young exosomes injected in aged mice. Int J Nanomedicine 2018, 13, 5335-5345, doi:10.2147/IJN.S170680.
  11. Cao, H.; Yue, Z.; Gao, H.; Chen, C.; Cui, K.; Zhang, K.; Cheng, Y.; Shao, G.; Kong, D.; Li, Z.; et al. In Vivo Real-Time Imaging of Extracellular Vesicles in Liver Regeneration via Aggregation-Induced Emission Luminogens. ACS Nano 2019, 13, 3522-3533, doi:10.1021/acsnano.8b09776.
  12. Perets, N.; Betzer, O.; Shapira, R.; Brenstein, S.; Angel, A.; Sadan, T.; Ashery, U.; Popovtzer, R.; Offen, D. Golden Exosomes Selectively Target Brain Pathologies in Neurodegenerative and Neurodevelopmental Disorders. Nano Lett 2019, 19, 3422-3431, doi:10.1021/acs.nanolett.8b04148.
  13. Barbosa, M.; Gomes, C.; Sequeira, C.; Goncalves-Ribeiro, J.; Pina, C.C.; Carvalho, L.A.; Moreira, R.; Vaz, S.H.; Vaz, A.R.; Brites, D. Recovery of Depleted miR-146a in ALS Cortical Astrocytes Reverts Cell Aberrancies and Prevents Paracrine Pathogenicity on Microglia and Motor Neurons. Front Cell Dev Biol 2021, 9, 634355, doi:10.3389/fcell.2021.634355.
  14. Gomes, C.; Sequeira, C.; Likhite, S.; Dennys, C.N.; Kolb, S.J.; Shaw, P.J.; Vaz, A.R.; Kaspar, B.K.; Meyer, K.; Brites, D. Neurotoxic Astrocytes Directly Converted from Sporadic and Familial ALS Patient Fibroblasts Reveal Signature Diversities and miR-146a Theragnostic Potential in Specific Subtypes. Cells 2022, 11, doi:10.3390/cells11071186.

Reviewer 2 Report

This research tried to show the protective effect of the secretome from anti-miR-124 treated mSOD1 motor neurons.  Although the idea is interesting, their control group is not appropriate. I want to ask the authors to repeat experiments with appropriate controls.

Major comments

1.     “WT mice + vehicle” and “mSOD1 mice + vehicle” can not be controls for “mSOD1 mice + anti-miR-124-MN secretome”. Either “mSOD1 mice + secretome from mSOD1-NSC-34 cell” or “mSOD1 mice + secretome from WT SOD1-NSC-34 cell” will be controls.

2.     Authors need to show the differences between “secretome from anti-miR-124-MN mSOD1-NSC-34 cell” and “secretome from normal mSOD1-NSC-34 cell”. Especially the level of miR-124 level.

3.     From figures 3 to 10, the authors compared two groups each. These are not reader-friendly. The authors need to compare three groups in one graph.

4.     Figure 9AB and figure 9 legend are not displayed in pdf file. 

Minor comments

1.     A notation of “anti-miR-124-MN secretome” changes a lot. For example, anti-miR-124 (Fig3), secretome from anti-miR-124-treated mSOD1 motor neurons (fig4). Please use consistent notation.

2.     Introduction part: Authors use many words which mean the time (lately, recently, very recently). This makes it hard to read.   Line 48, reference 3 must be incorrect, please check.   Line 58 “may” <= what is the subject in this sentence?   Line 88, authors should not break the line.   Around lines 88-96, the authors described in detail the experiment methods. It can be shorter and simpler.

3.     In figure 2 and line337~338, the authors showed the presence of EVs 72 hours after injection. Is this still EVs? Since authors stained the lipid layer of EVs with a “fluorescence linker kit”, the shown GFP fluorescence might be just lipid or GFP itself. It is difficult to believe that EVs are stable for such long time inside the cells.   

        DOI: 10.1002/advs.202003505

4.     Figure 9 G and I, Y-axis ranges are bad.

5.     Line 106. Mistakes

6.     Line 110, “After 24 h and to” mistake? 

Reviewer 3 Report

The manuscript addresses the therapeutic potentials of the anti-miR-124-mSOD1-motor neuron (MN) secretome in preventing disabilities in the mSOD1 mice. Following an intrathecal injection in mSOD1 mice at 12 weeks of age, the authors observed: (i) improvement in motor disabilities in the mSOD1 mice; (ii)  prevention of muscle loss; (iii) halting of MN and myelin loss; (iv)  control of pathological glial mechanisms.

They propose using secretome as a therapeutic tool to be tested in stratified patients with an up-regulation of miR-124 at the symptomatic onset. 

I would just revise it for some typos 

I like this work carefully performed and properly discussed 

Author Response

We are grateful for the Reviewer appreciation of our study. We edited and corrected the manuscript for some typos. Similarly, we rephrased some sentences for clarity in the revised version.

Round 2

Reviewer 1 Report

The authors have responded to all my comments adequately.

Author Response

We are pleased that the reviewer considers that we have responded to all the  comments adequately.

Thank you so much.

Reviewer 2 Report

I couldn't convince myself to read their answer to my first comment.
However, 
I found that authors compared the effects of secretomes ± anti-miR-124 treatment in their previous paper (reference #13) figure 5.
After this, I could understand their idea.
Authors should describe this result in the introduction around line 83.

minor comment
line 123
"Anti-miRTM 124 inhibitor " maybe incorrect.

This revised version was very difficult to read. The figures are poorly positioned. Authors need to check carefully before submission. Also, the microsoft tracking history has too much junk information. This is the first time I have seen such a revised paper. Yellow highlighting is good enough.

Author Response

Comments and Suggestions for Authors

I couldn't convince myself to read their answer to my first comment.
However, I found that authors compared the effects of secretomes ± anti-miR-124 treatment in their previous paper (reference #13) figure 5.
After this, I could understand their idea.
Authors should describe this result in the introduction around line 83.

We addressed this issue by rephrasing the sentences (lines 72-82) as it follows:

  • miRNA mimics and inhibitors have been proposed as therapeutics to modulate dysregulated miRNAs in cancer and multiple sclerosis [15,16]. Lately, we demonstrated that the incubation of the secretome from anti-miR-124-treated mSOD1 MNs (preconditioned secretome) was able to preserve the increased levels of IL-1β, IL-18, HMGB1, arginase 1 and iNOS in the microglia treated with the mSOD1 secretome [13]. Interestingly, our data evidenced that the targeting of miR-124 toward normal values in ALS MNs enriched in miR-124 abrogates miR-125b overexpression and causes miR-146a/miR-21 downregulation in both cells and secretome, thus inhibiting a pathological inflamma-miRNA profile in mSOD1 MNs. Such preconditioned secretome also showed similar benefits in the spinal organotypic cultures from the mSOD1 mice, by preventing the dysregulation of inflammation-associated genes and cell demise.

minor comment
line 123
"Anti-miRTM 124 inhibitor " maybe incorrect.

We understand the reviewer’s comment, but it is not incorrect. Please see indication in the following link: https://www.thermofisher.com/order/genome-database/details/mirna/AM10691

  • To better clarify, it is now indicated as “anti-miR-124 (Ambion® Anti-miRTM miRNA inhibitor, #AM10691)” in the revised version.

This revised version was very difficult to read. The figures are poorly positioned. Authors need to check carefully before submission. Also, the microsoft tracking history has too much junk information. This is the first time I have seen such a revised paper. Yellow highlighting is good enough.

We apologize for such junk information. We usually use yellow highlighting, but we should have misunderstood the Editor request of using the “Track Changes” function.

Considering the request of this Reviewer we have highlighted in yellow the sentences that were modified in the Revised Manuscript second round in Word. The clean pdf manuscript is also submitted.

Thank you so much for the criticisms and suggestions, which improved the present version of the manuscript.